# Alchemist: Turning Public Text-to-Image Data into Generative Gold

**Valerii Startsev***
Yandex Research, HSE

**Alexander Ustyuzhanin***
Yandex

**Alexey Kirillov**
Yandex, MSU

**Dmitry Baranchuk**
Yandex Research

**Sergey Kastryulin**
Yandex Research

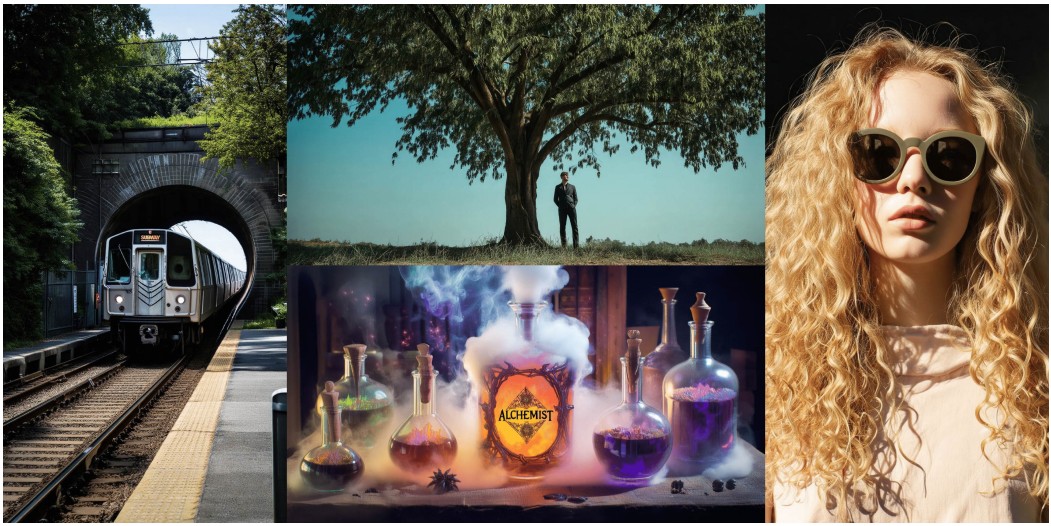

Figure 1: Images generated by Stable Diffusion 3.5 Large fine-tuned on Alchemist, demonstrating enhanced aesthetic quality and complexity while maintaining prompt adherence.

## Abstract

Pre-training equips text-to-image (T2I) models with broad world knowledge, but this alone is often insufficient to achieve high aesthetic quality and alignment. Consequently, supervised fine-tuning (SFT) is crucial for further refinement. However, its effectiveness highly depends on the quality of the fine-tuning dataset. Existing public SFT datasets frequently target narrow domains (e.g., anime or specific art styles), and the creation of high-quality, general-purpose SFT datasets remains a significant challenge. Current curation methods are often costly and struggle to identify truly impactful samples. This challenge is further complicated by the scarcity of public general-purpose datasets, as leading models often rely on large, proprietary, and poorly documented internal data, hindering broader research progress. This paper introduces a novel methodology for creating general-purpose SFT datasets by leveraging a pre-trained generative model as an estimator of high-impact training samples. We apply this methodology to construct and release **Alchemist**, a compact (3,350 samples) yet highly effective SFT dataset. Experiments demonstrate that Alchemist substantially improves the generative quality of five public T2I models while preserving diversity and style. Additionally, we release the fine-tuned models' weights to the public

---

*Equal contribution.

39th Conference on Neural Information Processing Systems (NeurIPS 2025) Track on Datasets and Benchmarks.

# 1   Introduction

Generative text-to-image (T2I) models, such as DALL-E 3 [1], Imagen 3 [2], and Stable Diffusion 3 [3], have demonstrated remarkable advancements in synthesizing high-fidelity and diverse images from textual descriptions. These models, typically pre-trained on vast internet-scale datasets, are applied in creative industries, for content generation, and in scientific visualizations. Despite their capabilities, the continuous pursuit of enhanced generative quality and better alignment with user intent remains a central research focus.

Supervised Fine-Tuning (SFT) has become a vital technique for adapting these pre-trained foundation models, whether to specialize them for particular domains or aesthetics, or to broadly elevate their general generative performance. However, the success of SFT is critically dependent on the quality and composition of the fine-tuning dataset. Current practices for SFT dataset curation often rely on extensive manual human selection. This process is not only costly and challenging to scale but can also be surprisingly ineffective. The specific characteristics of text-image pairs that render a sample "good" for SFT – that is, likely to maximally boost general model quality – are frequently subtle, not obvious, and difficult for humans to consistently verbalize or identify. Alternative approaches, such as filtering large web datasets with simple heuristics or employing synthetic data generation, have their own limitations in efficiently targeting high-impact samples or ensuring quality and diversity without introducing new biases.

These methodological challenges are compounded by a significant scarcity of publicly available, general-purpose SFT datasets explicitly designed to broadly enhance T2I models. While numerous domain-specific fine-tuning datasets exist, they serve niche applications rather than general quality improvement. Furthermore, several recent state-of-the-art models (e.g., Emu [4], PixArt-$\alpha$ [5], Kolors [6], SANA [7], YaART [8]) report using internal datasets for their SFT stages. These datasets, however, remain closed-source and are often described with insufficient detail in publications, severely limiting the research community's ability to replicate findings, understand their construction principles, or develop comparable open resources. This lack of accessible, well-characterized, general-purpose SFT datasets impedes broader progress in systematically improving T2I models.

To address these challenges, we propose a novel approach that leverages the intrinsic understanding of a pre-trained generative model to more effectively guide the SFT dataset creation process. Our core idea is that a pre-trained generative model can itself serve as an estimator of data quality, pinpointing samples most likely to contribute positively to the fine-tuning objective and maximize generative improvements in downstream models. To demonstrate the practical utility of this methodology, we created the Alchemist dataset and subsequently used it to fine-tune five publicly available text-to-image models, the improved weights of which we release as part of our contributions.

This work aims to provide the first open general-purpose alternative to proprietary fine-tuning pipelines, enabling reproducible research and commercial applications.

We present the following contributions:

- A principled methodology for curating high-quality, general-purpose SFT datasets by leveraging a pre-trained generative model to identify samples that maximize post-SFT model improvement.
- Alchemist, a compact (3,350 samples) yet highly effective SFT dataset constructed via our methodology, significantly enhances text-to-image generation quality while maintaining output diversity and style.
- Open-sourced, fine-tuned weights for five publicly available text-to-image models, demonstrating performance gains over their baselines after SFT with Alchemist.

The remainder of this paper details our methodology for dataset creation, presents experimental results showcasing its effectiveness, and discusses the outcomes and limitations of our research.

# 2   Related Work

**Supervised Fine-Tuning of Text-to-Image Models.**   Early text-to-image models like DALL-E [9] and Latent Diffusion Models (LDMs) [10] were primarily pre-trained on vast, uncurated web-scale datasets (e.g., LAION-5B [11]), focusing on general generative capabilities without specific SFT

stages. A significant advancement came with Emu [4], which demonstrated that an SFT stage on a smaller, high-quality, curated dataset substantially improved instruction following and aesthetic quality. Subsequently, SFT or similar refinement stages became standard in state-of-the-art models. For instance, PixArt-$\alpha$ [5] enhances outputs through training on data with higher aesthetic quality, boosting the training efficiency. Later works [6, 7, 8] also employ multi-stage training including fine-tuning on aesthetically filtered data. This trend highlights crucial role of SFT in achieving high-quality and controllable image generation.

**Supervised Fine-Tuning Datasets for Text-to-Image Models.** Publicly available, general-purpose SFT datasets for text-to-image models remain limited. LAION-Aesthetics [12], derived from LAION-5B [11] by filtering for predicted aesthetic scores, is widely used. However, its quality is often considered inferior compared to closed source datasets. While more recent efforts, such as LAION-Aesthetics V2 [13], aim to improve upon this, a meticulously verified, general-purpose public SFT dataset is largely absent. In contrast, domain-specific SFT datasets are more common, such as the Danbooru dataset [14] for anime-style generation and WikiArt dataset [15] for classical and modern art generation. These datasets achieve strong performance within their specific domains but typically at the cost of the model's broader generative abilities, causing it to overfit to the narrow domain of the SFT data. The scarcity of high-quality, general-purpose public SFT datasets motivates our work.

**Quality Assessment of Text-to-Image Models.** Evaluating text-to-image generation quality is complex. Automated metrics like Fréchet Inception Distance (FID) [16] and Inception Score (IS) [17], while common, often correlate poorly with human perception [18, 19]. Consequently, more comprehensive evaluations rely on human assessment. Studies for models like Imagen [20] and Parti [21], for example, involved human raters evaluating photorealism, text-image alignment, absence of artifacts, and compositionality. Standardized prompt sets such as DrawBench [20] and T2I-CompBench [22] facilitate structured comparison. In this work we provide some automated metrics while building main conclusions based on carefully designed multi-aspect human-based evaluations.

## 3 Dataset Formation

Our goal is to create a general-purpose supervised fine-tuning dataset capable of significantly enhancing the generative quality of pre-trained text-to-image (T2I) models while preserving their diversity in content, composition, and style. To achieve this, we introduce a multi-stage filtering pipeline designed to create a small set of exceptionally high-quality samples from a vast pool of uncurated internet data. A core principle of our methodology involves leveraging a pre-trained diffusion model as a sophisticated estimator in the final filtering stage to identify text-image pairs with the highest potential to boost downstream SFT performance. This section details our pipeline, the effectiveness of which is demonstrated in Section 4.

**Overview.** The dataset construction process starts from a vast, diverse pool of $\mathcal{O}(10$ billion$)$ images aggregated from web-scraped sources. Some dataset curation pipelines for image-text models discussed in literature [23] impose text-based filtering at the initial stages, discarding samples with poorly structured, noisy or semantically misaligned captions. While this approach mitigates low-quality training pairs, we argue that it is increasingly restrictive given recent advances in multi-modal captioning models. Early text filtering eliminates potentially valuable visual content that could be re-captioned with synthetic texts. Instead, we compose our data curation pipeline as purely image-based. The relatively high-quality set of data that passes first filtering stages is further filtered using diffusion-based sample quality estimator and then captioned with a Vision-Language Model (VLM) to obtain the final SFT dataset. Figure 2 provides an overview of the dataset formation pipeline.

**Stage 1: Foundational Safety and Resolution Filtering.** The first filtering stage addresses basic image requirements. We discarded images identified as containing Not-Safe-For-Work (NSFW) content through an automated classifier. Subsequently, we applied a resolution filter, retaining only those images with an area exceeding $1024 \times 1024$ px. This step ensures that the candidate pool consists of sufficiently high-resolution and safe visual content for subsequent processing.

**Stage 2: Coarse-Grained Quality Assessment and Filtering.** Following the foundational filters, we employed a suite of lightweight binary classifiers for rapid, coarse-grained quality assessment.

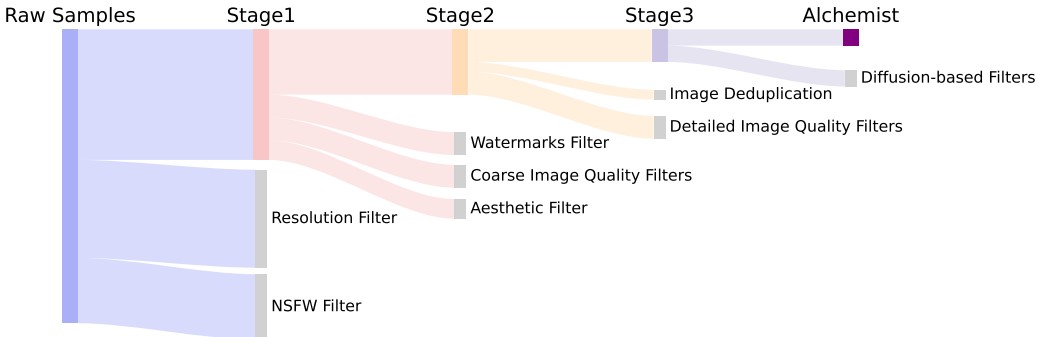

Figure 2: Overview of the multi-stage image filtering pipeline. Beginning with a web-scale collection of raw data, the pipeline sequentially filters images to isolate a high-quality subset optimally suited for supervised fine-tuning of text-to-image models.

Following the foundational filters, we employed a suite of lightweight binary classifiers for rapid, coarse-grained quality assessment. These classifiers were trained to identify and remove images exhibiting severe degradation, prominent watermarks, noticeable compression artifacts, significant motion blur, or low aesthetic appeal. The classifiers for general quality and aesthetics were trained on public Image Quality Assessment (IQA) [24, 25] and Image Aesthetics Assessment (IAA) [26, 27] datasets. For watermark detection, we utilized a proprietary classifier built on a Vision Transformer (ViT) backbone, trained on a large, diverse internal dataset to ensure high reliability. Classification thresholds were manually calibrated to aggressively remove the worst-quality examples. The first two stages significantly reduced the dataset size, yielding approximately one billion images for further processing.

**Stage 3: Deduplication and Fine-Grained Quality Refinement.**  With a more manageable dataset size, we applied more computationally intensive methods. First, to enhance visual diversity, we performed image deduplication by computing SIFT-like local features [28], clustering images by similarity, and retaining only one representative (highest preliminary quality score) per cluster. [2] Second, for fine-grained perceptual quality assessment, we utilized the TOPIQ no-reference IQA model [29]. The distribution of TOPIQ scores in our pre-filtered data was uni-modal, peaking around 0.6. Our selection of a stringent threshold of $> 0.71$ was a deliberate, data-driven decision aimed at balancing two competing objectives: maximizing technical image quality and preserving broad domain diversity. Through empirical ablation studies, we identified a clear trade-off: thresholds higher than $0.71$ introduced significant content bias by disproportionately selecting for narrow domains (e.g., architecture and interiors), while lower thresholds allowed a higher rate of images with subtle artifacts that negatively impacted downstream SFT performance. The $0.71$ threshold thus represented an optimal balance, effectively isolating images with minimal distortions and artifacts while maintaining the broad thematic coverage crucial for a general-purpose dataset. This filtering stage resulted in approximately 300 million high-quality images.

**Perceptual and Compositional Refinement.**  The objective here is to find a subset of images with a rare combination of visual characteristics such as high aesthetic quality, optimal color balance, and substantial image complexity that are hypothesized to maximize SFT quality. Existing IQA and IAA models often struggle to holistically capture this specific blend of attributes crucial for SFT.

Our hypothesis is that a pre-trained diffusion model, through its learned representations, inherently encodes these desired characteristics, particularly within its cross-attention mechanisms which mediate text-image alignment during generation. To leverage this, we developed a scoring function based on cross-attention activations. We utilize a long, multi-keyword prompt designed to evoke the target visual qualities (e.g., including terms like "high quality", "artistic", "aesthetic", "complex"). For each image, we extract cross-attention activation norms corresponding to these keywords. To identify the most discriminative activations, we manually scored a calibration set of 1,000 images based on the aforementioned SFT-desirable criteria, forming "higher-quality" and "lower-quality"

---

[2]Our choice of a SIFT-based deduplication method over alternatives was driven by its favorable trade-off between performance and computational cost at scale. We empirically verified its effectiveness to be was sufficient for our goals.

---
**Algorithm 1:** Diffusion-based Quality Estimator
---
**Input:** $X_{HQ}, X_{LQ}$: Two groups of train images of higher and lower visual quality

$X$: Test images, $|X| = N$

$\epsilon_\theta$: Pretrained text-to-image generative model

$\mathcal{P}$: Predefined prompt with tokens $\{w_1, ..., w_M\}$

$L$: Number of cross-attention layers

$K$: Number of top discriminative features

$t$: Timestep for activation extraction

**Output:** Quality scores $\mathbf{f} \in \mathbb{R}^N$

1. **Extract activations:**

**for** *each image $x \in \mathbb{R}^{h \times w}$ in $X_{HQ} \cup X_{LQ} \cup X$* **do**

    Save cross-attn maps $\{A_{l,m}^{(x)} \in \mathbb{R}^{h_l \times w_l}\}_{\substack{l=1...L \\ m=1...M}}$ during noise prediction via $\epsilon_\theta(x, \mathcal{P}, t)$

    Compute spatial activation norms:

        $N_{l,m}^{(x)} = \|A_{l,m,:,:}^{(x)}\|_2 \quad \forall l \in \{1, ..., L\}, m \in \{1, ..., M\}$

**end**

2. **Find (layer, token) pairs with most discriminative features:**

**for** *each $(l, m)$ pair* **do**

    $s_{l,m} \leftarrow 0$

    **for** *each $(x_{HQ} \in X_{HQ}, x_{LQ} \in X_{LQ})$ pair* **do**

        Compute separation score:

        $s_{l,m} \mathrel{+}= \mathbb{I}[N_{l,m}^{(x_{HQ})} > N_{l,m}^{(x_{LQ})}]$

    **end**

**end**

Select top-$K$ $(l, m)$ pairs with highest $s_{l,m}$: $\mathcal{K} = \{(l_1, m_1), ..., (l_K, m_K)\}$

3. **Compute scores:**

**for** *each image $x \in X$* **do**

    $\mathbf{f}_x = \sum_{(l,m)\in\mathcal{K}} N_{l,m}^{(x)}$

**end**

**return** *Quality scores* $\mathbf{f}$

---

groups. We then identified the top-$K$ activation indices that best separated these two groups. The final score for any given image is an aggregation (summation) of its activation norms at these top-$K$ indices (details in Algorithm 1). A detailed discussion of methodological details, including prompt engineering and choice of inference timestep $t$, is provided in Appendix C.

Using this diffusion-based scoring function, we evaluated all $N \approx 300$ million images from Stage 3 and selected the top-$n$ samples. The SFT dataset size ($n$) is a critical hyperparameter. Through ablation studies (detailed in Section 4.4), we determined that $n = 3,350$ provides best model quality improvements with no observable loss of generative diversity.

**Final Re-captioning and the Alchemist Dataset.** The 3,350 images curated by our pipeline, though visually exceptional, retained their original, often noisy, web captions. Effective supervised fine-tuning (SFT) necessitates appropriate textual guidance. Our preliminary studies highlighted the importance of caption style, finding that captions resembling moderately descriptive user-like prompts rather than exhaustively detailed ones achieve better SFT results. Therefore, we re-captioned the entire set using a proprietary image captioning model tuned to produce such user-centric descriptions. This re-captioning ensured consistent and relevant textual pairings. The resulting Alchemist dataset consists of these 3,350 refined image-text pairs, used for subsequent analysis and SFT.

## 4 Experiments

We empirically evaluate the effectiveness of **Alchemist** as an SFT dataset for open-source Stable Diffusion (SD) models. Our goal is to verify whether a compact, highly curated dataset like Alchemist can significantly boost image generation quality and outperform LAION-Aesthetics v2 [13] as a standard publicly available SFT alternative. Below we discuss experimental setup and present results of fine-tuning with Alchemist in terms of human-perceived generation quality and common automated metrics.

### 4.1 Experimental Setup

**Models and Datasets.** We evaluate our proposed methodology across five widely-used pre-trained text-to-image models based on Stable Diffusion: SD1.5[3], SD2.1[4], SDXL1.0[5], SD3.5 Medium[6], and SD3.5 Large[7] [10, 30, 3]. For each base model, we utilize the official checkpoints and Diffusers-based [31] publicly available fine-tuning code to establish three comparison points:

- **Baseline:** The original official model weights;

- **Alchemist-tuned:** The baseline model fine-tuned on our proposed Alchemist dataset (comprising 3,350 samples);

- **LAION-tuned:** The baseline model fine-tuned on a size-matched subset (3,350 samples) drawn from the LAION-Aesthetics v2 dataset [13], specifically selecting samples with aesthetic scores $>= 6.5$. This serves as a control to assess the effectiveness of Alchemist compared to a standard, high-aesthetics filtered dataset of equivalent size. We additionally ablate dataset size for LAION in Appendix E.2.

**Fine-Tuning and Hyperparameter Selection.** We employed a full fine-tuning approach, updating all parameters of the base models. To identify optimal settings for each (model, dataset) combination, we conducted a grid search over key hyperparameters, including learning rate, EMA momentum, and the number of training steps. The specific search ranges and the final selected hyperparameters for each configuration are detailed in Appendix F.1. Checkpoint selection and early stopping decisions during this tuning process were guided by performance on a dedicated validation set. This validation set consisted of 128 prompts selected from the PartiPrompts benchmark [21], following the methodology employed in SD3 [3].

**Test Set for Final Evaluation.** The final performance assessment of the best checkpoints selected via the validation process was conducted on a separate, unseen test set. This test set comprised 500 distinct prompts also drawn from PartiPrompts [21], ensuring no overlap with the prompts used during validation or hyperparameter tuning.

Additionally, to further mitigate the prompt distribution leak into the models final assessment and ensure generalizability we conducted extra human side-by-side comparisons with prompts from DrawBench [20]. The results are detailed in Appendix B.

### 4.2 Evaluation Protocol

**Human Side-by-Side Evaluation** Our primary method for evaluating model performance relies on human perception via side-by-side (SbS) comparisons. For each comparison pair (e.g., Alchemist-tuned vs. Baseline), we generated images using prompts from the validation or test sets; detailed parameters are provided in Appendix G). Three expert annotators were independently presented with the generated images. Annotators evaluated the pairs based on four criteria:

- **Image-Text Relevance:** Accuracy of the image content relative to the text prompt;

- **Aesthetic Quality:** Overall visual appeal, including composition and style;

- **Image Complexity:** Richness of detail and content within the scene;

- **Fidelity:** Presence and severity of defects, artifacts, distortions, or undesirable elements.

For each criterion, annotators selected the preferred image, with the option of indicating a tie. The final outcome for a given prompt and criterion was determined by majority vote among the three annotators. We assess the statistical significance of the aggregate win rates using a two-sided binomial test ($p < 0.05$). Details regarding the SbS interface and instructions are provided in Appendix H.

---

[3] https://huggingface.co/stable-diffusion-v1-5/stable-diffusion-v1-5
[4] https://huggingface.co/stabilityai/stable-diffusion-2-1
[5] https://huggingface.co/stabilityai/stable-diffusion-xl-base-1.0
[6] https://huggingface.co/stabilityai/stable-diffusion-3.5-medium
[7] https://huggingface.co/stabilityai/stable-diffusion-3.5-large

| Model | Side-by-Side Win Rate | | | | VLM Win Rate | | Automatic Metrics ($\Delta$) | | | |
|---|---|---|---|---|---|---|---|---|---|---|
| | Rel.↑ | Aes. ↑ | Comp. ↑ | Fidel. ↑ | SC ↑ | PQ ↑ | $FD_{DINOv2}$ ↓ | CLIP ↑ | IR ↑ | HPS-v2 ↑ |
| **SD1.5-Alchemist** | | | | | | | 129.8 | 0.277 | **0.38** | **0.270** |
| vs baseline | 0.53 | 0.64 | 0.78 | 0.47 | 0.52 | 0.60 | 131.5 | 0.279 | 0.02 | 0.243 |
| vs LAION-tuned | 0.47 | 0.60 | 0.73 | 0.45 | | | **112.1** | **0.286** | 0.32 | 0.260 |
| **SD2.1-Alchemist** | | | | | | | **95.6** | 0.281 | 0.62 | **0.282** |
| vs baseline | 0.57 | 0.69 | 0.81 | 0.56 | 0.52 | 0.62 | 129.3 | 0.276 | 0.18 | 0.253 |
| vs LAION-tuned | 0.49 | 0.56 | 0.72 | 0.52 | | | 112.4 | **0.287** | **0.65** | 0.278 |
| **SDXL-Alchemist** | | | | | | | 97.4 | 0.286 | 0.76 | **0.292** |
| vs baseline | 0.52 | 0.61 | 0.78 | 0.51 | 0.50 | 0.50 | **73.4** | 0.293 | 0.71 | 0.283 |
| vs LAION-tuned | 0.49 | 0.58 | 0.78 | 0.57 | | | 108.9 | **0.294** | **0.81** | 0.291 |
| **SD3.5M-Alchemist** | | | | | | | **76.2** | 0.286 | **1.07** | **0.295** |
| vs baseline | 0.51 | 0.57 | 0.67 | 0.50 | 0.52 | 0.52 | 81.4 | **0.287** | 0.97 | 0.292 |
| vs LAION-tuned | 0.48 | 0.58 | 0.73 | 0.49 | | | 87.9 | 0.286 | 0.87 | 0.274 |
| **SD3.5L-Alchemist** | | | | | | | **80.9** | 0.287 | **1.12** | **0.299** |
| vs baseline | 0.49 | 0.62 | 0.72 | 0.41 | 0.5 | 0.52 | 91.4 | 0.286 | 1.01 | 0.298 |
| vs LAION-tuned | 0.47 | 0.57 | 0.76 | 0.55 | | | 91.1 | **0.297** | 1.10 | 0.294 |

Table 1: Comparison of Alchemist-tuned models, baselines, and LAION-Aesthetics-tuned models. The table reports human and VLM win rates (by aspect) w.r.t. Alchemist-tuned models and automated metrics values for each model variant. Green indicates statistically significant improvement ($p < 0.05$), gray no statistically significant change, and red a statistically significant decline. For automated metrics **bold** means the best value among three model variants.

**VLM-as-a-judge Evaluation.** To provide a greater reliability of evaluation we used currently popular VLM-as-a-judge approach to assess the finetuned models generation quality in comparison to corresponding baselines. Specifically, we computed a text-to-image variant of VIEScore [32], using GPT-4o [33] as the backbone due to its demonstrated strong correlation with human judgments. VIEScore assesses two primary dimensions: Semantic Consistency (SC), which aligns with our **Relevance** criterion, and Perceptual Quality (PQ), which most closely corresponds to our **Fidelity** criterion.

For each model pair and prompt, we used GPT-4o to determine a winner, loser, or tie based on these two dimensions and analyzed results with the same statistical methodology as our human evaluations.

**Automated Metrics** To complement human judgments, we report established automated metrics. These include **FD-DINOv2**, which calculates the Fréchet Distance [16] using DINOv2 [34] features, and **CLIP Score** [35], based on ViT-L/14 [36] image-text similarity. Additionally, we employ learned human preference predictors: **ImageReward** (IR) [37] and **HPS-v2** [38]. All automated metrics were computed on the standard MJHQ-30K dataset [39].

## 4.3 Results

**Human Evaluation Results.** The results from human side-by-side (SbS) evaluations demonstrate how fine-tuning impacts the four specified assessment criteria. Regarding **Image-Text Relevance**, fine-tuning with Alchemist did not yield statistically significant differences compared to either the baseline or the LAION-tuned models across most tested architectures ($p > 0.05$). This indicates that the improvements observed in other aspects do not compromise prompt fidelity.

Conversely, Alchemist fine-tuning demonstrated substantial and statistically significant improvements in both **Aesthetic Quality** and **Image Complexity**. Compared to the respective baseline models, Alchemist-tuned versions achieved human preference win rates up to 20% higher. Furthermore, Alchemist consistently outperformed the size-matched LAION-Aesthetics-tuned variants on these two criteria, with win rate advantages ranging from +12% to +20% across the different base models.

In terms of **Fidelity**, the results were mixed. While many models showed no significant change, fine-tuning with Alchemist led to a marginal but statistically significant decrease in perceived fidelity for certain architectures (average win rate decrease of approximately 5% against baseline in those cases). We hypothesize this may represent a tradeoff associated with generating more complex and detailed images, a point further discussed in Sections 5 and 6.

**VLM-as-a-judge Evaluation Results.** The results of VLM-based evaluation presented in the Table 1 are broadly consistent with our human assessments for the corresponding criteria (**Relevance** and **Fidelity**). The VLM-judge confirms the preservation of textual relevance and, interestingly, assesses our model's fidelity more favorably than our expert human annotators. This provides strong evidence that modern VLMs can serve as a reliable and scalable proxy for evaluating objective aspects of T2I model performance. However, we also note that current VLM evaluation frameworks, including VIEScore, are primarily designed to assess more objective criteria like prompt alignment and artifact detection. Reliably capturing more subjective and nuanced human preferences, such as **Aesthetic Appeal** and **Image Complexity**, remains a significant challenge and an important direction for future research in the field [32, 40].

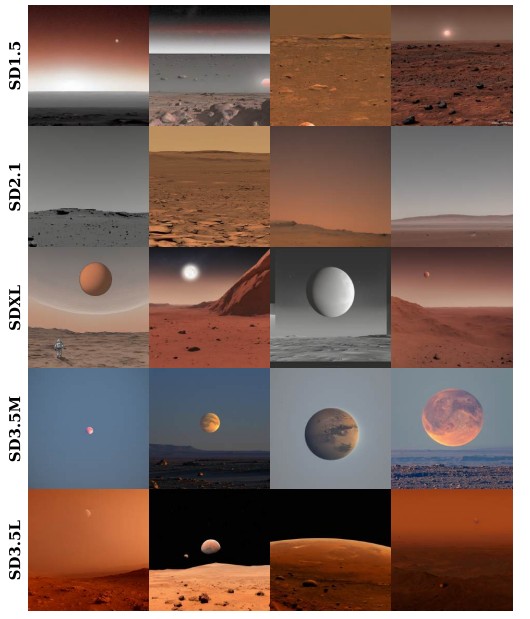 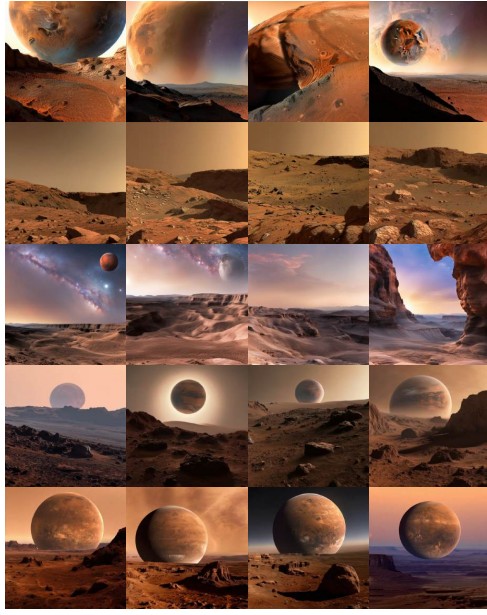

(a) Baseline.  (b) Alchemist-tuned.

Figure 3: Examples of images generated by five Stable Diffusion models for the prompt "*Mars rises on the horizon.*" **before** and **after** tuning on Alchemist.

**Qualitative Analysis** To visually complement these quantitative assessments and human judgments, Figure 3 presents qualitative examples of images generated by several models, showcasing outputs from their baseline versions alongside those after fine-tuning with Alchemist. These visual comparisons directly illustrate the enhancements in aesthetic appeal, detail, and overall image complexity reported above. The examples also suggest that fine-tuning with Alchemist does not lead to a noticeable decline in the diversity of content or stylistic range generated by the models. A more extensive collection of qualitative results, including additional model comparisons and prompt examples, is provided in Appendix J.

**Automated Metric Results.** These findings from human evaluations and qualitative analysis are further confirmed by automated metrics. Improvements in FD-DINOv2, CLIP Score, and the learned preference scores (ImageReward, HPS-v2) were observed for most models after fine-tuning with Alchemist, particularly when compared to the untuned baselines (see Table 1 for detailed results). The comparison against LAION-Aesthetics-tuned models on these metrics also generally favored the Alchemist variants, supporting the conclusions drawn from human preferences. The modest improvements shown by automated metrics underscores their limitation in capturing the nuanced perceptual quality that our human evaluation has successfully revealed.

## 4.4  Dataset Size Ablation

To assess the impact of strict filtering, we created two larger Alchemist variants (approx. 7k and 19k samples) by relaxing the selection threshold of our diffusion-based quality estimator. These

datasets inherently contained samples with lower diffusion-guided quality scores than the original 3,350-sample Alchemist. We then fine-tuned all five base models on these 7k and 19k variants.

As summarized in Table 4, fine-tuning on both larger datasets yielded consistently inferior performance across all models compared to the compact 3,350-sample Alchemist. An additional, dedicated hyperparameter sweep for the 7k and 19k datasets confirmed this finding, as no configuration achieved quality comparable to that of the original Alchemist. These results underscore that exceptional sample quality curated by strict, diffusion-guided filtering is more critical for maximizing SFT efficacy than sheer dataset volume.

We further investigate the interesting question of smaller Alchemist sizes in Appendix E.3.

## 5  Discussion

Fine-tuning with Alchemist substantially enhances aesthetic quality and image complexity across diverse Stable Diffusion models, highlighting the power of targeted SFT with compact, high-impact datasets. Our findings, however, also prompt further discussion.

| Model | Side-by-Side Win Rate | | | |
|---|---|---|---|---|
| | Rel.↑ | Aes. ↑ | Comp. ↑ | Fidel. ↑ |
| **SD1.5-Alchemist-3k** | | | | |
| vs Alchemist-7k | 0.44 | 0.62 | 0.64 | 0.47 |
| vs Alchemist-19k | 0.43 | 0.62 | 0.67 | 0.49 |
| **SD2.1-Alchemist-3k** | | | | |
| vs Alchemist-7k | 0.45 | 0.61 | 0.62 | 0.55 |
| vs Alchemist-19k | 0.46 | 0.60 | 0.76 | 0.53 |
| **SDXL-Alchemist-3k** | | | | |
| vs Alchemist-7k | 0.49 | 0.61 | 0.66 | 0.57 |
| vs Alchemist-19k | 0.48 | 0.65 | 0.73 | 0.53 |
| **SD3.5M-Alchemist-3k** | | | | |
| vs Alchemist-7k | 0.48 | 0.61 | 0.58 | 0.53 |
| vs Alchemist-19k | 0.50 | 0.75 | 0.81 | 0.58 |
| **SD3.5L-Alchemist-3k** | | | | |
| vs Alchemist-7k | 0.54 | 0.52 | 0.47 | 0.55 |
| vs Alchemist-19k | 0.52 | 0.68 | 0.70 | 0.57 |

Figure 4: Comparison of models fine-tuned on Alchemist variants of different sizes. The table reports human win rates (by aspect) of Alchemist-3k-tuned models against models tuned on 7k and 19k variants of Alchemist. Green indicates statistically significant improvement ($p < 0.05$), gray no significant change, and red a statistically significant decline.

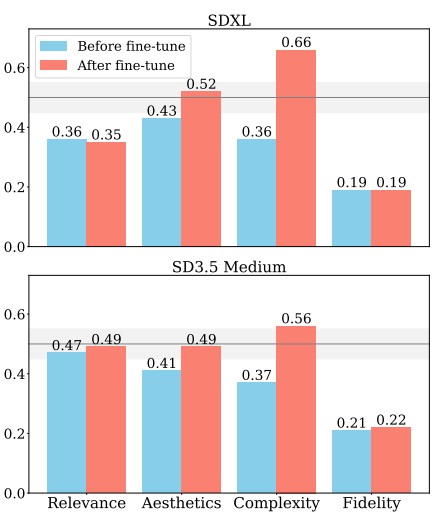

Figure 5: Results of SbS comparison of SDXL, SD3.5 Medium before and after fine-tuning versus FLUX. Grey shaded region shows the interval of statistical insignificance.

We observe that Alchemist fine-tuning yields varied improvements and trade-offs across models. Notably, later architectures like SD3.5 showed a slight decrease in fidelity, a trend less apparent in earlier models. This difference likely stems from the base models' histories: newer models may have already incorporated some variation of fine-tuning after their initial pre-training. Consequently, our general-purpose SFT with Alchemist, while beneficial, might introduce characteristics that subtly conflict with these existing, highly specific optimizations, leading to the observed fidelity trade-off. Earlier models, with less such prior refinement, may more readily absorb Alchemist's broad quality enhancements.

We also observe an inherent link between increased image complexity and a potential drop in fidelity. Guiding models towards richer scenes, a strength Alchemist confers, inherently provides more opportunities for minor artifacts. This suggests that achieving high complexity and maximal fidelity may necessitate techniques beyond general SFT.

Furthermore, our results confirm this SFT approach minimally impacts image-text relevance. This aspect seemingly depends more on model architecture, initial pre-training data, and dedicated alignment methods, rather than fine-tuning primarily focused on visual style.

Ultimately, Alchemist's quality improvements effectively bridge the performance gap between traditional SD models and cutting-edge solutions. Figure 5 reveals that Alchemist-tuned SDXL and SD3.5 Medium exhibit aesthetic quality and image complexity comparable to leading models like FLUX.1-dev [41] despite having 4 times less parameters. This underscores that data-efficient SFT on well-pre-trained foundations remains a viable path to significant quality advancements.

## 6 Limitations

While Alchemist fine-tuning significantly enhances image aesthetics and complexity (Section 4.3), two primary limitations warrant acknowledgment. Firstly, this pursuit of visual richness can introduce a marginal decrease in perceived fidelity for some models, a trade-off more pronounced in highly optimized later architectures (e.g., SDXL, SD3.5) than in earlier ones (e.g., SD1.5, SD2.1) which showed clearer net quality gains without substantial defect increases. This suggests that pushing already high-performing models towards greater complexity via SFT may inherently surface minor imperfections. Secondly, our approach did not yield significant improvements in image-text relevance. This aspect appears to be more dependent on factors like model architecture, initial pre-training, and dedicated alignment techniques, rather than the visual quality-focused SFT employed here. Despite these points, Alchemist effectively achieves its primary goal of elevating key visual qualities in text-to-image models using a compact, targeted dataset.

## 7 Conclusion

This work introduced Alchemist, a compact (3,350 samples) supervised fine-tuning (SFT) dataset, and its novel creation methodology leveraging a pre-trained diffusion model as a key quality estimator, followed by re-captioning with moderately descriptive, user-like prompts. Extensive experiments across five Stable Diffusion models demonstrated Alchemist's effectiveness in significantly boosting aesthetic quality and image complexity, outperforming baselines and a size-matched LAION-Aesthetics SFT. While image-text relevance remained largely unaffected and a minor complexity-fidelity trade-off emerged for highly optimized models, ablation studies underscored the crucial role of our strict filtering and compact dataset size for achieving superior SFT outcomes. Our principled, data-efficient approach and the public release of the Alchemist dataset and fine-tuned model weights offer valuable resources and insights for advancing text-to-image generation through high-quality SFT.

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

## Appendices

Here we show additional results on generation of non-square images (Appendix A), DrawBench evaluation (Appendix B), detail dataset collection procedure (Appendix C), investigate the impact on generated image diversity (Appendix D), provide additional ablations of data filtering, LAION-Aesthetics size selection, smaller Alchemist dataset sizes, model architecture and captioning approach (Appendix E), detail our experimental and inference settings (Appendices F and G), describe and provide examples of human evaluation (Appendix H). In conclusion we discuss broader impact (Appendix I) and provide additional visualizations for qualitative assessment (Appendix J).

## A    Non-square Aspect Ratio Generation

In the early era of diffusion-based text-to-image generation, models such as SD1.5 and SD2.1 were trained exclusively on square images, limiting their ability to generate images with different aspect ratios. As diffusion technology advanced, the concept of bucketed training was introduced [42]. This approach organizes training batches by resolution, where each batch contains images of identical resoluitons, but the image size varies between training iterations. This methodology enabled models to generate images across diverse aspect ratios.

The Alchemist dataset comprises images with varying aspect ratios, facilitating bucketed fine-tuning. This training approach ensures the model can produce high-quality images beyond the traditional square format.

In our experiments with SDXL, SD3.5 Medium, and SD3.5 Large, we implemented bucketing for the latent representations, varying the latent resolution across batches. In addition to evaluating square image generation, we present side-by-side (SbS) comparisons between Alchemist-tuned versions of SDXL, SD3.5 Medium, and SD3.5 Large against their original counterparts, which were inherently designed to support multi-aspect ratio image generation.

| Model | Side-by-Side Win Rate | | | |
|---|---|---|---|---|
| | Rel.$\uparrow$ | Aes. $\uparrow$ | Comp. $\uparrow$ | Fidel. $\uparrow$ |
| **SDXL-Alchemist**$_{[h,w]}$ | | | | |
| vs baseline$_{[1280,768]}$ | 0.53 | 0.62 | 0.78 | 0.49 |
| vs baseline$_{[896,1152]}$ | 0.48 | 0.63 | 0.83 | 0.49 |
| **SD3.5M-Alchemist**$_{[h,w]}$ | | | | |
| vs baseline$_{[1280,768]}$ | 0.52 | 0.55 | 0.65 | 0.50 |
| vs baseline$_{[896,1152]}$ | 0.50 | 0.60 | 0.70 | 0.51 |
| **SD3.5L-Alchemist**$_{[h,w]}$ | | | | |
| vs baseline$_{[1280,768]}$ | 0.51 | 0.60 | 0.71 | 0.40 |
| vs baseline$_{[896,1152]}$ | 0.51 | 0.67 | 0.72 | 0.40 |

Table 2: We evaluated the ability of Alchemist-tuned models and baseline models to generate images with non-square aspect ratios. For each model, we produced images of resolution $[h, w]$, where the exact values of $h$ and $w$ are specified in the baseline subscripts. The table reports human win rates (by aspect) w.r.t. Alchemist-tuned models. Green indicates statistically significant improvement ($p < 0.05$), gray no statistically significant change, and red a statistically significant decline.

The side-by-side (SbS) comparison results align with those in Table 1: fine-tuning enhances the aesthetic quality and complexity of generated images without sacrificing text coherence, though for SD3.5 Large it occasionally introduces more artifacts due to increased detail. This confirms that the dataset enables generation of images of various aspect ratios without compromising their quality compared to the baseline square size.

# B  DrawBench Evaluation Results

Evaluating the model generation quality on diverse prompt distributions is crucial for validating the general-purpose nature of our SFT dataset. To address this, we conducted an additional evaluation using the challenging DrawBench benchmark. We performed the same side-by-side (SbS) human evaluation, comparing our Alchemist-tuned models against their baselines on the DrawBench prompt set. The results are summarized in the Table 3.

| Model | Side-by-Side Win Rate | | | |
| --- | --- | --- | --- | --- |
| | Rel.↑ | Aes. ↑ | Comp. ↑ | Fidel. ↑ |
| **SD1.5-Alchemist** vs baseline | 0.52 | 0.66 | 0.77 | 0.54 |
| **SD2.1-Alchemist** vs baseline | 0.53 | 0.72 | 0.85 | 0.58 |
| **SDXL-Alchemist** vs baseline | 0.53 | 0.60 | 0.74 | 0.57 |
| **SD3.5M-Alchemist** vs baseline | 0.52 | 0.67 | 0.74 | 0.53 |
| **SD3.5L-Alchemist** vs baseline | 0.53 | 0.73 | 0.80 | 0.41 |

Table 3: Comparison of Alchemist-tuned models and their baselines conducted on DrawBench prompt set. The table reports human win rates (by aspect) w.r.t. Alchemist-tuned models. Green indicates statistically significant improvement ($p < 0.05$), gray no statistically significant change, and red a statistically significant decline.

As the results demonstrate, the performance gains from Alchemist fine-tuning are not specific to the PartiPrompts distribution. The Alchemist-tuned models maintain their statistically significant advantage in both Aesthetic Quality and Image Complexity on DrawBench. This provides strong evidence that Alchemist imparts a fundamental improvement to generative quality that generalizes well across different and more challenging prompt scenarios.

# C  Dataset Collection Details

## C.1  Timestep Selection

The timestep $t \in [0.0, 1.0]$ in the input in Algorithm 1 is a crucial parameter of our approach. When $t$ approaches $0.0$, the generated image is almost fully formed, and the influence of the text prompt diminishes significantly. Conversely, as $t$ approaches $1.0$, the activations become dominated by noise and lose interpretability. Through empirical analysis, we identified $t = 0.25$ as an optimal balance point and employed this value across all binary classifiers.

## C.2  Diffusion-based Estimator Prompt

Another critical element of Algorithm 1 is its input text prompt $\mathcal{P}$. We define it as follows:

*"complex. detailed. simple. bokeh effect. abstract. photorealistic. artistic. stylized. aesthetic. cinematic. instagram filters. color correction. midjourney. ugly. distorted. blurry. rendering. AI-generated. synthetic. high quality. low quality. pixelated. low illumination."*

This prompt formulation integrates both empirical findings and theoretical principles of visual appeal, specifically targeting perceptual factors that influence human judgments of image quality. The template incorporates descriptors that capture both desirable and undesirable attributes across key visual dimensions:

1. **Image complexity.** Our experimental analysis revealed that images with minimal visual complexity (e.g., images with monochrome backgrounds or reduced detail density) contributed negligibly to model generation quality and are being overshadowed by more intricate, information-rich counterparts. Furthermore, while the inclusion of images featuring bokeh effects demonstrated a stabilizing influence on training dynamics, we observed a corresponding degradation in overall model performance. Consequently, our final curation pipeline excluded both minimalistic imagery and samples exhibiting excessive bokeh distortion.

2. **Art.** Artistic images and real life photos inherently differ in their visual characteristics and require separate processing pipelines. Photographic quality relies on objective technical parameters that are more suited for measurements, whereas artistic quality depends on subjective stylistic choices and are often out-of-domain for the most of classifiers. For these reasons we focus on incorporating such feature in our prompt.

3. **Aesthetic and Color correction.** We aim to estimate the aesthetic quality of images by learning discriminative features associated with coherent color palette, sharp focus on key subjects, satisfying photo composition rules and other properties of aesthetically compelling images from those that are commonly produced by amateur photography. A critical subtask in computational aesthetic evaluation involves assessing color fidelity, as a significant portion of consumer-grade photographs exhibit improper white balance, inaccurate saturation, or unnatural tonal distributions due to uncalibrated capture devices and lack of skill. This aspect specifically identifies images with professional-grade color correction characterized by balanced neutral tones, highlight-to-shadow transitions, proper color palette and saturation.

4. **Compression and noise.** Beyond aesthetic considerations, technical image quality presents a challenge for generative model training. Degradation categories, such as compression artifacts from JPEG and WebP formats, sensor-level noise and optical aberrations, affect high-frequency features learning that results with increase in image generation artifacts.

To mitigate bias from any single concept, we intentionally designed our diffusion-based estimator prompt to be long and diverse, not narrow. As detailed above, it incorporates a broad range of keywords associated with general visual quality ("high quality", "aesthetic", "complex", "photorealistic", etc.), rather than focusing on a specific, subjective style. The goal of this prompt is to activate a general-purpose "quality vector" within the diffusion model, not to steer the selection towards a niche aesthetic. While a direct ablation on multiple scoring prompts is computationally prohibitive as it would require re-scoring all 300 million candidate images and re-running all fine-tuning experiments for each prompt variation, we use our comprehensive downstream evaluations to indirectly validate our prompt choice. The results demonstrate that Alchemist-tuned models did not collapse or overfit:

- **Qualitative analysis** (Figure 3, Appendix J) shows preserved **stylistic and content diversity**.
- **Human and VLM SbS evaluations** and **CLIP Scores** (Table 1) confirmed that **Image-Text Relevance** was not degraded.
- **FID scores** (Table 1) remained stable, suggesting **no significant distributional shift** away from the baseline.

## D   Image Generation Diversity

We conducted an analysis of intra-prompt diversity, following the methodology in [43]. For each prompt in our test set, we generated $N = 4$ images using different seeds. We then computed the average pairwise cosine distance between the feature embeddings of these images (extracted using a CLIP ViT-L model [44]). A higher score indicates greater diversity. The results are presented in Table 4.

The results confirm a noticeable decrease in the diversity metric for Alchemist-tuned models. However, we posit that this reduction does not signify a loss of global stylistic or conceptual coverage. We believe, to some extent, these results reflect the model's increased reliability and its convergence towards high-quality outputs. Baseline models often exhibit higher "error diversity" by producing off-prompt, lower-quality, or nonsensical generations, which, while distant in feature space, do not represent a desirable creative range. Alchemist fine-tuning reduces this undesirable diversity by consistently generating high-fidelity images that are more thematically coherent with the prompt.

| Model | Diversity↑ | |
| --- | --- | --- |
| | Original | Alchemist-tuned |
| SD1.5 | 0.37 [0.36; 0.39] | 0.26 [0.25; 0.27] |
| SD2.1 | 0.34 [0.33; 0.35] | 0.20 [0.19; 0.21] |
| SDXL | 0.26 [0.25; 0.27 ] | 0.22 [0.21; 0.23] |
| SD3.5M | 0.22 [0.21; 0.23] | 0.18 [0.17; 0.19] |
| SD3.5L | 0.20 [0.19, 0.21] | 0.17 [0.16, 0.18] |

Table 4: Comparison of image generation diversity before and after tuning on Alchemist.

This interpretation is strongly supported by our other findings. A true collapse in diversity (i.e., mode collapse) would lead to a sharp decline in Image-Text Relevance and CLIP Scores [35], as the model would fail to generate a wide range of concepts accurately. As shown in our main results (Table 1), these metrics remained stable and robust after fine-tuning. This indicates that the model's ability to address diverse prompts is fully preserved, and the measured decrease in diversity is partially attributable to the elimination of low-quality failure modes.

## E    Additional Ablations

### E.1    Filtration Approach

This subsection examines the necessity of the diffusion-based estimator in our filtration pipeline. To evaluate its importance, we removed this component and implemented a more rigorous filtering process using TOPIQ-IAA [29] and classifiers trained on TAD-66k [26], KonIQ-10k [24] and IC-9600 [45]. This experiment is motivated by the fact that Image Quality and Aesthetics (IQA/IAA) models are a common tool for quality assessment, and recent work has even explored their direct integration into the generation process itself [46]. This approach maintained the same sample size as Alchemist while selecting for high aesthetic quality and substantial complexity. All other steps in our filtration pipeline remained unchanged.

We adhere to the same image appeal considerations detailed in Appendix C. Our pipeline begins with complexity filtering using the IC-9600 classifier, where we apply a lower threshold to exclude monochromatic or overly simplistic images.

Next, we employ aesthetic and image quality estimators trained on TAD-66K and KonIQ-10k correspondingly. Based on our analysis, the KonIQ-based classifier aligns more closely with human judgment for high-scoring images. Consequently, we apply a stricter threshold for KonIQ compared to the TAD-66K-based model, which shows less consistent performance for top-tier samples.

Following data filtration using non-diffusion-based estimators, we fine-tuned the baseline Stable Diffusion models referenced in our main text. We then evaluated these fine-tuned models through side-by-side (SbS) comparisons with their corresponding Alchemist-tuned counterparts. The results of this evaluation are presented in Table 5.

The newly obtained models exhibit two key limitations: (1) reduced image-text coherence and (2) reduced fidelity. We attribute these effects to several factors:

1. The IC-9600-trained classifier retains excessively complex images in its top selections, whereas our diffusion-based estimator effectively identifies samples with "moderate" complexity - a key characteristic for improving generation quality. Training on overly complex data consistently degrades output fidelity.

2. Overly strict thresholds on both TAD-66k and KonIQ-10k filters introduce significant content bias in the dataset, ultimately compromising text-to-image alignment during generation.

3. Visual analysis of the dataset, along with side-by-side (SbS) model comparisons after tuning on this data, shows an important limitation. Existing classifiers are not able to reliably tell apart average-quality images from the aesthetically outstanding samples needed for successful SFT.

| Model | Side-by-Side Win Rate | | | |
|---|---|---|---|---|
| | Rel.↑ | Aes. ↑ | Comp. ↑ | Fidel. ↑ |
| **SD1.5-Alchemist** vs IC9600-TAD66k-KonIQ-sorted | 0.78 | 0.54 | 0.52 | 0.62 |
| **SD2.1-Alchemist** vs IC9600-TAD66k-KonIQ-sorted | 0.80 | 0.53 | 0.59 | 0.68 |
| **SDXL-Alchemist** vs IC9600-TAD66k-KonIQ-sorted | 0.84 | 0.63 | 0.58 | 0.68 |
| **SD3.5M-Alchemist** vs IC9600-TAD66k-KonIQ-sorted | 0.94 | 0.46 | 0.46 | 0.79 |
| **SD3.5L-Alchemist** vs IC9600-TAD66k-KonIQ-sorted | 0.91 | 0.62 | 0.52 | 0.72 |

Table 5: Comparison of Alchemist-tuned models against models tuned on the dataset filtrated using TOPIQ-IAA and IC9600, TAD-66k and KonIQ-10k trained classifiers. The table reports human win rates (by aspect) w.r.t. Alchemist-tuned models. Green indicates statistically significant improvement ($p < 0.05$), gray no statistically significant change, and red a statistically significant decline.

This ablation study shows that using TOPIQ-IAA and classifiers trained on TAD-66k, KonIQ-10k and IC9600 does not lead to Alchemist-level integral quality of fine-tuned models.

### E.2 LAION-Aesthetics Size

In our primary analysis, we compared the 3,350-sample Alchemist dataset against an equally sized random subset of LAION-Aesthetics v2 [13] images meeting our minimum size threshold (area $\geq 1024 \times 1024$ px). In this subsection we validate that the sample size was not the reason for inferior performance of the LAION-based finetuning.

To ablate the influence of the dataset size we select a complete set of 31k samples from LAION-Aesthetics v2 that pass resolution-based selection. Consistently with our previous fine-tuning experiments, we performed a hyperparameter sweep to train the top-performing models for this dataset. After that, we conducted side-by-side (SbS) comparisons against Alchemist fine-tuned versions (Table 6).

| Model | Side-by-Side Win Rate | | | |
|---|---|---|---|---|
| | Rel.↑ | Aes. ↑ | Comp. ↑ | Fidel. ↑ |
| **SD1.5-Alchemist** vs full LAION-tuned | 0.55 | 0.59 | 0.77 | 0.54 |
| **SD2.1-Alchemist** vs full LAION-tuned | 0.54 | 0.62 | 0.76 | 0.63 |
| **SDXL-Alchemist** vs full LAION-tuned | 0.54 | 0.66 | 0.86 | 0.63 |
| **SD3.5M-Alchemist** vs full LAION-tuned | 0.55 | 0.65 | 0.82 | 0.52 |
| **SD3.5L-Alchemist** vs full LAION-tuned | 0.52 | 0.62 | 0.72 | 0.60 |

Table 6: Comparison of Alchemist-tuned models and models tuned on the full LAION-Aesthetics v2 dataset. The table reports human win rates (by aspect) w.r.t. Alchemist-tuned models. Green indicates statistically significant improvement ($p < 0.05$), gray no statistically significant change, and red a statistically significant decline.

Human evaluation results demonstrate that models trained on the full LAION-Aesthetics v2 dataset continue to underperform those fine-tuned with Alchemist, particularly in measures of aesthetic quality and image complexity.

### E.3    Smaller Alchemist dataset size

We conducted an additional ablation study on smaller Alchemist sizes. Namely, we created three smaller variants of Alchemist by taking the top 500, 1,000, and 1,500 samples as ranked by our diffusion-based estimator. We then fine-tuned two representative models, SDXL and SD3.5 Medium, on these smaller datasets. For these runs, we linearly scaled down the number of training steps while keeping all other hyperparameters consistent. We compared the models tuned on these smaller datasets against the model tuned on our original 3,350-sample Alchemist (Alchemist-3k). The results of our side-by-side human evaluation are presented in Table 7.

| Model | Side-by-Side Win Rate | | | |
|---|---|---|---|---|
| | Rel.↑ | Aes. ↑ | Comp. ↑ | Fidel. ↑ |
| **SDXL-Alchemist-1.5k** | | | | |
| vs SDXL-original | 0.53 | 0.60 | 0.80 | 0.53 |
| vs SDXL-Alchemist-3k | 0.50 | 0.49 | 0.46 | 0.57 |
| **SDXL-Alchemist-1k** | | | | |
| vs SDXL-original | 0.52 | 0.57 | 0.72 | 0.52 |
| vs SDXL-Alchemist-3k | 0.52 | 0.45 | 0.41 | 0.59 |
| **SDXL-Alchemist-500** | | | | |
| vs SDXL-original | 0.54 | 0.63 | 0.69 | 0.54 |
| vs SDXL-Alchemist-3k | 0.52 | 0.41 | 0.36 | 0.60 |
| **SD3.5M-Alchemist-1.5k** | | | | |
| vs SD3.5M-original | 0.51 | 0.68 | 0.74 | 0.47 |
| vs SD3.5M-Alchemist-3k | 0.51 | 0.58 | 0.57 | 0.45 |
| **SD3.5M-Alchemist-1k** | | | | |
| vs SD3.5M-original | 0.51 | 0.70 | 0.77 | 0.48 |
| vs SD3.5M-Alchemist-3k | 0.5 | 0.58 | 0.59 | 0.49 |
| **SD3.5M-Alchemist-500** | | | | |
| vs SD3.5M-original | 0.52 | 0.70 | 0.79 | 0.50 |
| vs SD3.5M-Alchemist-3k | 0.52 | 0.56 | 0.54 | 0.49 |

Table 7: Comparison of models tuned on smaller versions of Alchemist. Green indicates statistically significant improvement ($p < 0.05$), gray no statistically significant change, and red a statistically significant decline.

While the 500-sample and 1,000-sample variants still show significant improvements over the baseline models, their performance relative to the 3,350-sample set is inconsistent. For instance, while the SD3.5 Medium model fine-tuned on 1k samples shows a competitive or even slightly improved performance, the SDXL model exhibits a clear trade-off, with gains in one aspect (e.g., Fidelity) coming at the cost of others (e.g., Aesthetics, Complexity).

This inconsistency across different model architectures suggests that while very small, highly curated datasets can be potent, they may not offer the same level of robust, general-purpose improvement. A larger set like our 3,350-sample Alchemist appears to provide a more stable and well-rounded enhancement across diverse models.

Furthermore, we hypothesize that fine-tuning on extremely small datasets, while potentially effective on some primary metrics, may carry a higher risk of negatively impacting other, unmeasured qualities, such as a more significant drop in generative diversity or overfitting to the few concepts present in the small set.

## E.4 Captioning Strategy

The relationship between prompt length and SFT performance is a critical and nuanced area. To verify the effectiveness of medium length, user-like prompts used in Alchemist, we created a new version of the Alchemist dataset by re-captioning the 3,350 images using a state-of-the-art long-captioning model, Qwen-VL-Max [47]. As shown in Table 8, this resulted in captions that were, on average, significantly longer and more descriptive than our original user-like prompts.

| Metric | Alchemist in-house captioner | Qwen2.5 VL 72B |
|---|---|---|
| Average number of symbols | 148 | 600 |
| Average number of words | 27 | 100 |

Table 8: Comparison of prompts lengths generated by in-house and open source models.

We then fine-tuned two representative models, SDXL and SD3.5 Medium, on this new "Alchemist-Long-Caption" dataset and compared their performance against the models fine-tuned on our original Alchemist dataset using the same SbS evaluation protocol and test set.

To additionally assess how other captioning models affect the quality of our dataset, we have also recaptioned the Alchemist with InternVL2 26B [48] and tuned SDXL and SD3.5 Medium models on the recaptioned data.

| Model | Side-by-Side Win Rate | | | |
|---|---|---|---|---|
| | Rel.↑ | Aes. ↑ | Comp. ↑ | Fidel. ↑ |
| **SDXL-Alchemist-QwenVL** | | | | |
| vs SDXL-original | 0.55 | 0.57 | 0.71 | 0.52 |
| vs SDXL-Alchemist | 0.52 | 0.44 | 0.40 | 0.57 |
| **SDXL-Alchemist-InternVL** | | | | |
| vs SDXL-original | 0.53 | 0.56 | 0.71 | 0.51 |
| vs SDXL-Alchemist | 0.51 | 0.47 | 0.41 | 0.52 |
| **SD3.5M-Alchemist-QwenVL** | | | | |
| vs SD3.5M-original | 0.54 | 0.61 | 0.63 | 0.50 |
| vs SD3.5M-Alchemist | 0.51 | 0.48 | 0.44 | 0.46 |
| **SD3.5M-Alchemist-InternVL** | | | | |
| vs SD3.5M-original | 0.52 | 0.64 | 0.69 | 0.44 |
| vs SD3.5M-Alchemist | 0.53 | 0.49 | 0.47 | 0.43 |

Table 9: Comparison of models tuned on Alchemist re-captioned with Qwen2.5-VL 72B and Intern-VL2 26B model with their baseline versions and SFT on Alchemist. Green indicates statistically significant improvement ($p < 0.05$), gray no statistically significant change, and red a statistically significant decline.

The evaluation in Table 9 demonstrates that while fine-tuning on the long-caption dataset still improves over the baseline, it is less effective at boosting Aesthetic Quality and Image Complexity compared to our original Alchemist dataset with its moderately descriptive, user-like prompts. Interestingly, the longer captions did provide a slight advantage in Fidelity (fewer artifacts), which suggests more descriptive prompts may reduce certain types of errors.

While the new models don't quite match the aesthetic and complexity levels of the original Alchemist-tuned versions, using captions from open-source models still leads to significant quality improvements in generation.

## E.5 Model Architecture

Although Alchemist has demonstrated generalizability across diverse Stable Diffusion models - differing in backbone architecture, training objectives, size, and fine-tuning history - we recognize

strong community interest in multimodal autoregressive models. Consequently, we have also fine-tuned Bagel [49] on the Alchemist dataset. Our side-by-side comparison from Table 10 shows that the Alchemist-tuned Bagel outperforms the original model in generating more aesthetic and complex images - though with a slight trade-off in fidelity.

| Model | Side-by-Side Win Rate | | | |
|---|---|---|---|---|
| | Rel.↑ | Aes. ↑ | Comp. ↑ | Fidel. ↑ |
| **Bagel-Alchemist** | | | | |
| vs Bagel-original | 0.55 | 0.57 | 0.71 | 0.52 |
| vs SDXL-Alchemist | 0.49 | 0.58 | 0.76 | 0.42 |

Table 10: Comparison of Bagel tuned on Alchemist model with its baseline version. Green indicates statistically significant improvement ($p < 0.05$), gray no statistically significant change, and red a statistically significant decline.

# F   Experimental Setting

## F.1   Hyperparameter Sweep and Train Setting

| Model | Learning Rates | Iterations (thousands) | EMA $\beta$ |
|---|---|---|---|
| SD1.5 | [1e-5, 2.5e-5, 8e-5] | [2.5, 5, 7.5, 10, 12.5] | [0.999, 0.9999] |
| SD2.1 | [1e-5, 2.5e-5, 8e-5] | [2.5, 5, 7.5, 10, 12.5] | [0.999, 0.9999] |
| SDXL | [1e-5, 2.5e-5, 8e-5] | [5, 10, 15, 20] | [0.999, 0.9999] |
| SD3.5 M | [5e-6, 2.5e-5, 8e-5] | [20, 40, 60, 80] | [0.9999] |
| SD3.5 L | [1e-6, 5e-6, 2.5e-5] | [20, 40, 60] | [0.9999] |

Table 11: Hyperparameter grids during our sweep. The particular choices were made according to the community best practices as well as our computational and human resource constraints.

We performed training hyperparameter sweep according to the Table 11 with the resulting training setup presented in the Table 12. Total batch size of 80, AdamW optimizer [50], Adam betas $\beta_1 = 0.9, \beta_2 = 0.999$ and constant learning rate scheduler were set the same for all the models. We didn't use learning rate warm-up. See Figure 6 for the training dynamics across different learning rates.

| Models | Learning Rate | Iterations | Weight Decay | EMA $\beta$ | GPUs | Mixed Precision |
|---|---|---|---|---|---|---|
| SD1.5 | 8e-5 | 5k | 1e-2 | 0.999 | 4 | `float16` |
| SD2.1 | 2.5e-5 | 7.5k | 1e-2 | 0.999 | 4 | `float16` |
| SDXL | 2.5e-5 | 10k | 1e-2 | 0.999 | 8 | `float16` |
| SD3.5 M | 5e-6 | 40k | 1e-4 | 0.9999 | 8 | `bfloat16` |
| SD3.5 L | 5e-6 | 20k | 1e-4 | 0.9999 | 8 | `bfloat16` |

Table 12: Final setup for fine-tuning on the Alchemist.

We used NVIDIA A100 with 80Gb of VRAM, PyTorch 2.6.0 [51], and CUDA 12.6. We varied the number of GPUs from 4 to 8 to ensure the total batch size of 80 on the one hand, and minimize the quantity of GPUs on the other. Distributed communication is performed via Open MPI [52]. We adopt Fully Sharded Data Parallel [53] for parameter and optimizer state sharding to reduce memory consumption and allow working with larger batch sizes.

Ultimately, with the final tuning setup it takes 12 GPU-hours to train SD1.5 model, 18 GPU-hours to train SD2.1, 80 GPU-hours to train SDXL, 480 GPU-hours to train SD3.5 Medium and 576 GPU-hours to train SD3.5 Large.

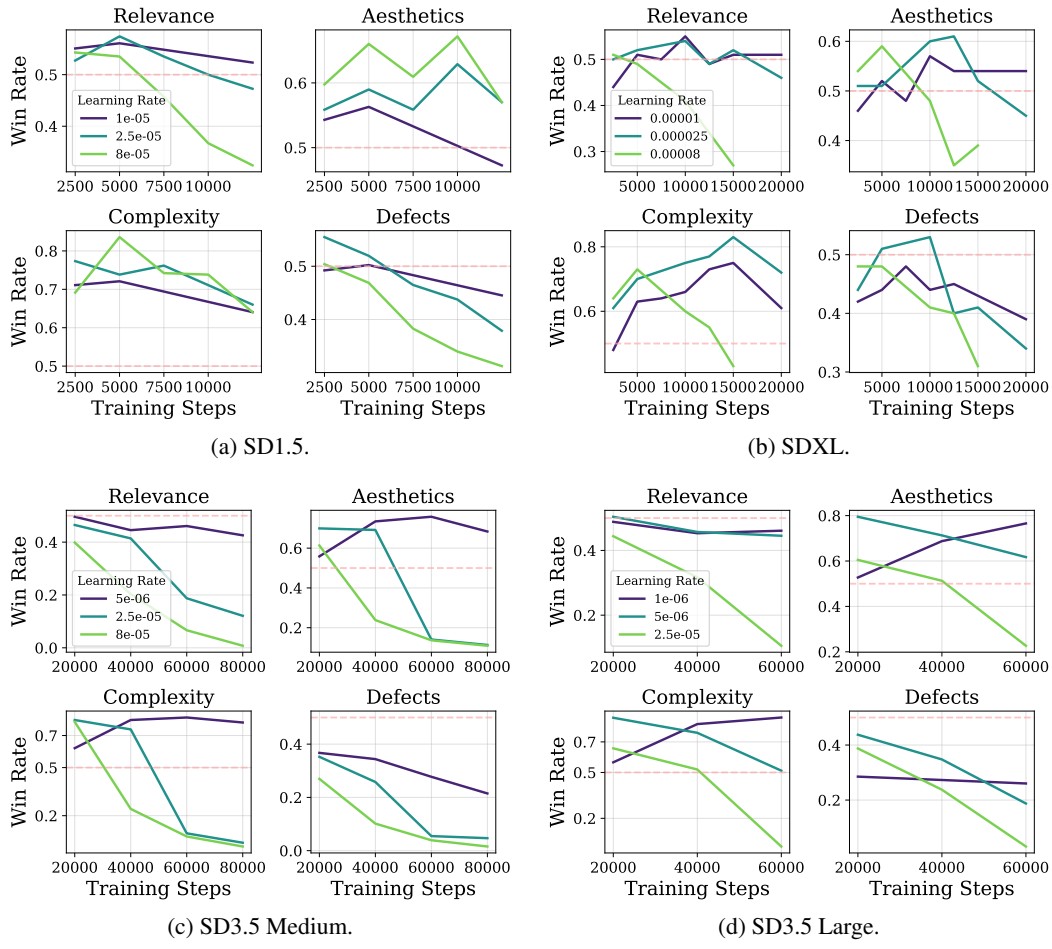

Figure 6: Training dynamics of SD models while tuning on Alchemist. We chose final checkpoints as those maximizing Aesthetics and Complexity while not reaching statistically significant decline in other aspects (if possible at all).

# G Inference Parameters

## G.1 Evaluation Setting

For all models except for the SD3.5 Large we conducted all inference measurements on 1 NVIDIA A100 GPU with 40GB of VRAM, batch size 1, using PyTorch 2.6.0 [51], and CUDA 12.6. For the SD3.5 Large we used the same software, but NVIDIA A100 GPU with 80GB of VRAM.

To generate images we used the parameters either recommended in the corresponding models' HuggingFace repositories or default ones from the Diffusers library. These parameters are provided in the Table 13. For SDXL we used 80/20% split of denoising steps between base and refiner models.

| Models | Guidance Scale | Number of Steps | Precision |
|--------|----------------|-----------------|-----------|
| SD1.5  | 7.5 | 50 | `float16` |
| SD2.1  | 7.5 | 50 | `float16` |
| SDXL   | 5.0 | 50 | `float16` |
| SD3.5 M | 4.5 | 40 | `bfloat16` |
| SD3.5 L | 3.5 | 28 | `bfloat16` |

Table 13: Inference parameters used in our work.

## G.2 Inference Parameter Sweep

Although all experiments used default inference parameters, we additionally evaluated model performance across different guidance scales and denoising steps. Due to limitations in human evaluation resources, we employed automated assessment using the ImageReward [37] metric for this analysis.

Varying guidance scale in $\in [1.0, 2.0, 4.0, 7.5]$ and number of inference steps in $\in [16, 32, 64]$, we show the dynamics of ImageReward in Figure 7. Consistent with our primary analysis, all metrics were computed on the MJHQ-30k benchmark dataset [39].

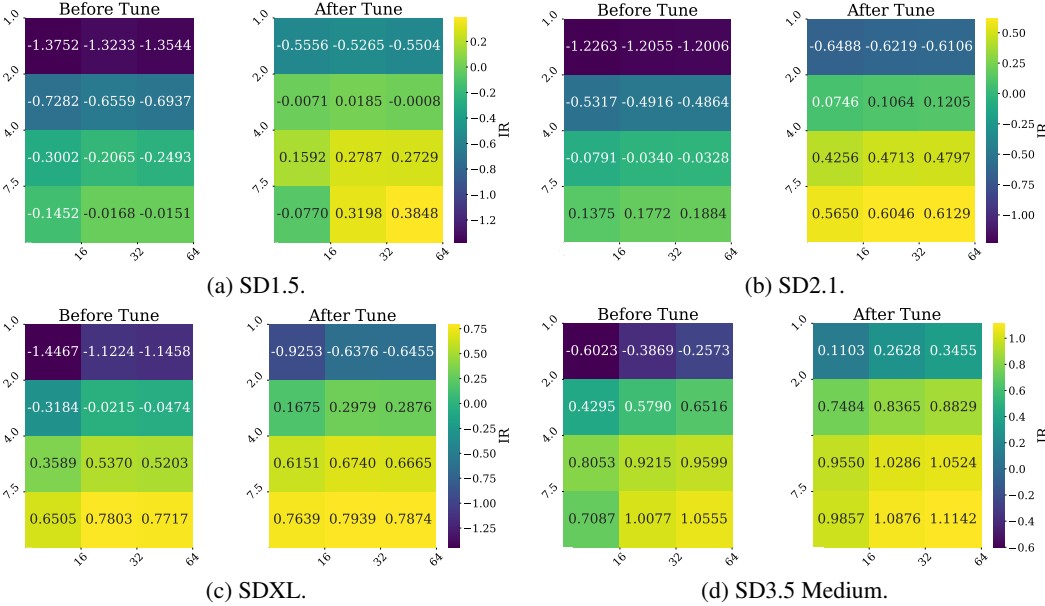

Figure 7: ImageReward metric change depending on guidance scale and number of denoising steps **before** and **after** tuning on Alchemist.

Our evaluations demonstrate that Alchemist-based tuning improves overall generation quality, evidenced by increased minimum and maximum ImageReward values. However, while the parameter heatmaps reveal ImageReward's preference for higher guidance scales, we caution against using these results as definitive optimization criteria. Prior work has established that excessive guidance scales induce overexposure artifacts in generated images [54], suggesting potential limitations in this metric's alignment with human perceptual quality.

## H Human Evaluation

We evaluated text-to-image generation quality through controlled side-by-side (SbS) comparisons, where professional assessors selected the superior image for each prompt-image pair. To ensure a fair and robust human preference study, all evaluations are conducted by a pool of more than 1000 expert annotators, with each image pair assessed by a randomly selected triplet of experts (assigning three annotators per pair is a well-established practice in the literature [55, 56]), with final judgments determined by majority vote.

Our evaluation team consists of trained professionals employed under ethical working conditions, including competitive compensation and risk disclosure. Assessors have received detailed and fine-grained instructions for each evaluation aspect and passed training and testing before accessing the main tasks. We highlight that our organizational equivalent of IRB approved the study.

Annotators evaluated a pair of images generated given the same validation or test prompt based on four criteria:

- **Image-Text Relevance:** Accuracy of the image content relative to the text prompt;
- **Aesthetic Quality:** Overall visual appeal, including composition and style;

- **Image Complexity:** Richness of detail and content within the scene;
- **Fidelity:** Presence and severity of defects, artifacts, distortions, or undesirable elements.

We provide the platform's interface during each aspect assessment in Figures 8,9,,10,11.

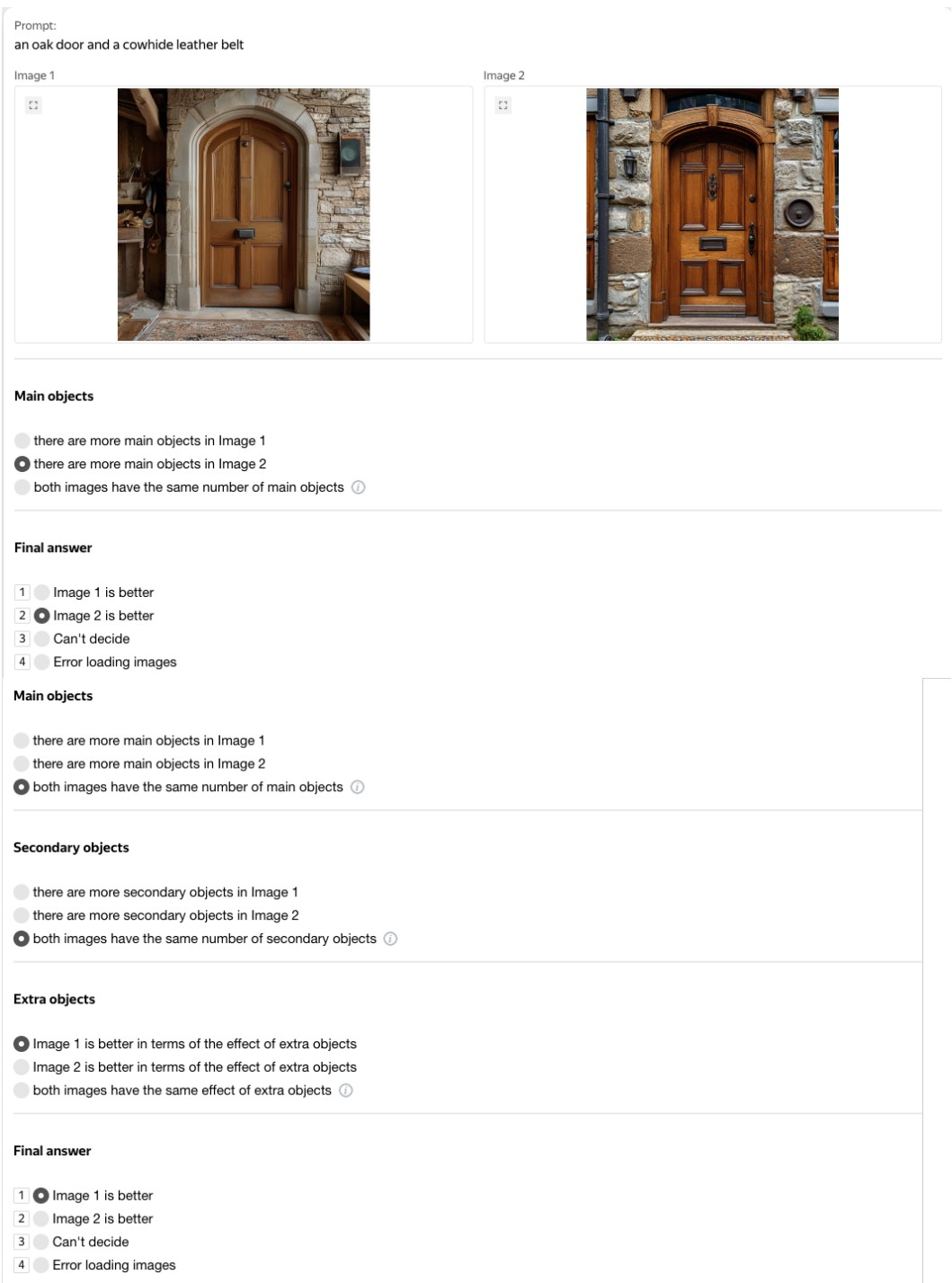

Figure 8: An example of a user interface for the **Image-Text Relevance** aspect of Human Evaluation with Side-by-Side comparisons.

From a mathematical point of view, human evaluation is a statistical hypothesis test. In particular, we are using a two-sided binomial test and its implementation from scipy [57] library to test the

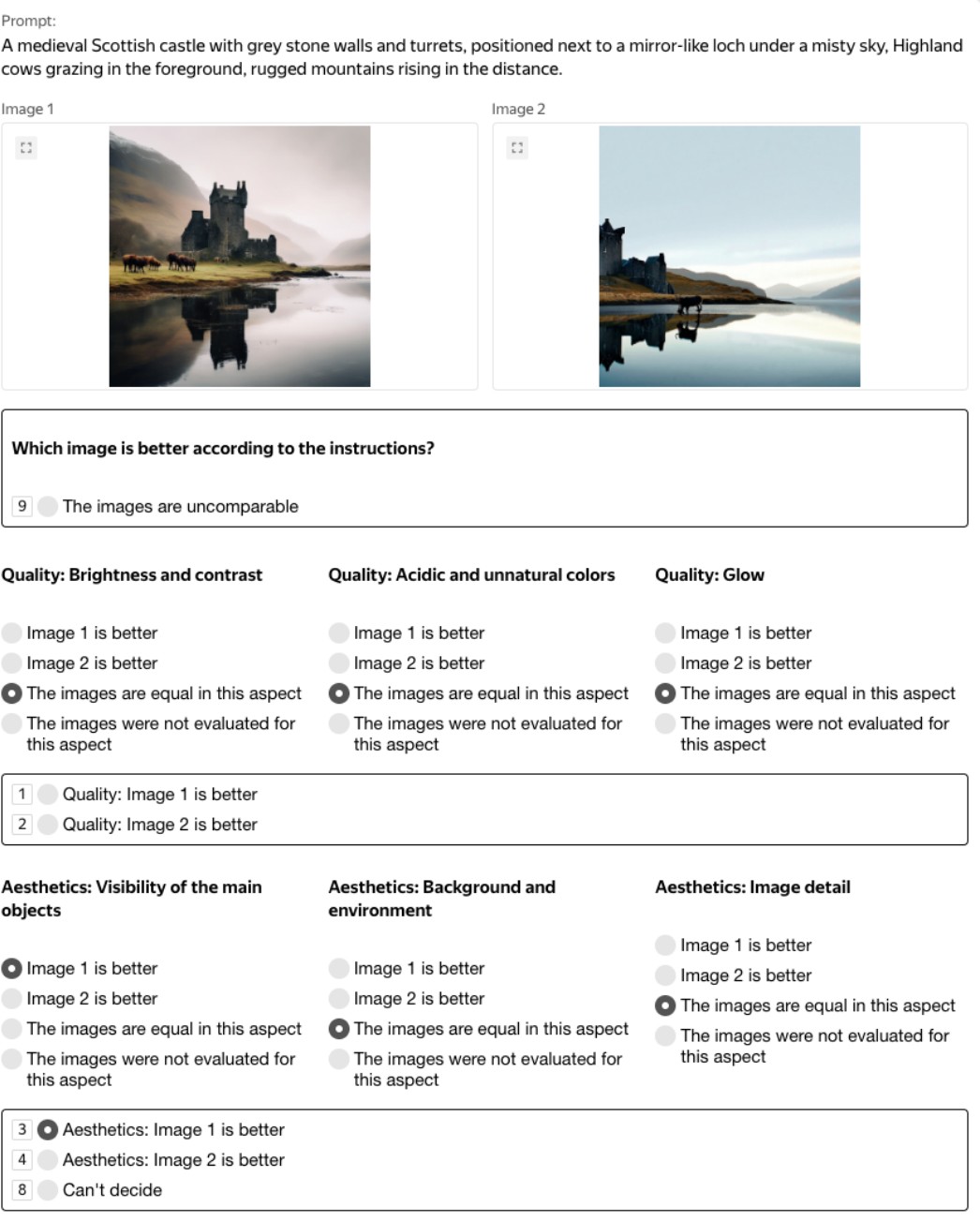

Figure 9: An example of a user interface for the **Aesthetics** aspect of Human Evaluation with Side-by-Side comparisons.

Prompt:
an oak door and a cowhide leather belt

Image 1

Image 2

**Defects in composition and watermarks**

○ Image 1 is better
○ Image 2 is better
○ Images are equal ⓘ

**Images style**

○ The images have the same style
○ The images differ in style / The images are uncomparable due to the style ⓘ
○ The verdict was based on the previous steps

**Defects of the main objects**

○ Image 1 is better
○ Image 2 is better
○ Can't decide
○ The verdict was based on the previous steps

**Defects of the secondary objects**

○ Image 1 is better
○ Image 2 is better
○ Can't decide
○ The verdict was based on the previous steps

**Final answer**

1 ○ Image 1 is better
2 ○ Image 2 is better
3 ○ Can't decide
4 ○ The images are uncomparable
5 ○ Error loading images

Figure 10: An example of a user interface for the **Fidelity** aspect of Human Evaluation with Side-by-Side comparisons.

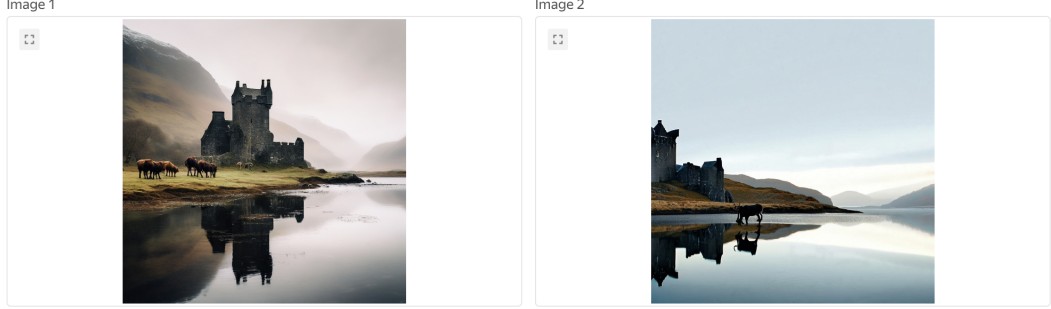

**Which image is more complex according to the instructions?**

1 ◯ Image 1 is better
2 ◯ Image 2 is better
8 ◯ Can't decide

Figure 11: An example of a user interface for the **Image Complexity** aspect of Human Evaluation with Side-by-Side comparisons.

null hypothesis of whether the two given models are equal in terms of image generation quality in 4 aspects independently. More precisely, for each aspect we calculate p-value as following:

```python
from scipy.stats import binomtest
# cnt_wins_baseline   - number of wins for baseline model
# cnt_wins_experiment - number of wins for experimental model
# cnt_equals          - number of equals
p_value  = binomtest(
    cnt_wins_baseline + cnt_equals / 2,
    cnt_wins_baseline + cnt_equals + cnt_wins_experiment
)
```

We reject the null hypothesis if is less than $0.05$, *i.e.*, at the 5% significance level.

# I  Broader Impact

The release of our open-source SFT dataset and fine-tuned text-to-image diffusion models carries significant societal implications, both positive and challenging. By openly sharing these resources, we aim to advance research in generative AI while fostering accessibility and reproducibility. The improved aesthetic quality and image complexity offered by our models can empower artists, educators, and small-scale creators, democratizing access to high-quality visual generation tools.

However, like all generative AI systems, these models present risks that must be carefully managed. The potential for misuse-such as generating deceptive imagery or deepfakes-necessitates safeguards, including provenance tracking and responsible deployment practices. The environmental impact of training and deploying such models also warrants consideration, encouraging the adoption of efficient fine-tuning techniques and shared computational resources.

To maximize the benefits of this work while mitigating risks, we emphasize the importance of transparency, collaboration, and oversight. Users should disclose AI-generated content where ethically relevant, and developers should engage with diverse stakeholders-including artists and ethicists-to ensure alignment with societal values. By proactively addressing these challenges, we hope to contribute to the responsible advancement of generative AI, ensuring that its benefits are widely accessible while minimizing unintended harm.

## J  More Visualizations

We provide more examples of images generated by models before and after fine-tune on Alchemist. The corresponding prompts are listed prior to the grids of images.

**Figure 12 prompts**

1. *"the Beatles crossing Abbey road"*
2. *"a portrait of a statue of the Egyptian god Anubis wearing aviator goggles, white t-shirt and leather jacket, flying over the city of Mars."*
3. *"Downtown LA at sunrise. detailed ink wash."*
4. *"a bird standing on a stick"*
5. *"a tornado passing over a corn field"*
6. *"a tennis court with tennis balls scattered all over it"*
7. *"a cloud in the shape of a castle"*
8. *"a flower with large red petals growing on the moon's surface"*
9. *"a diagram of brain function"*
10. *"a frustrated child"*
11. *"a woman with long black hair and dark skin"*
12. *"a macro photograph of brain coral"*

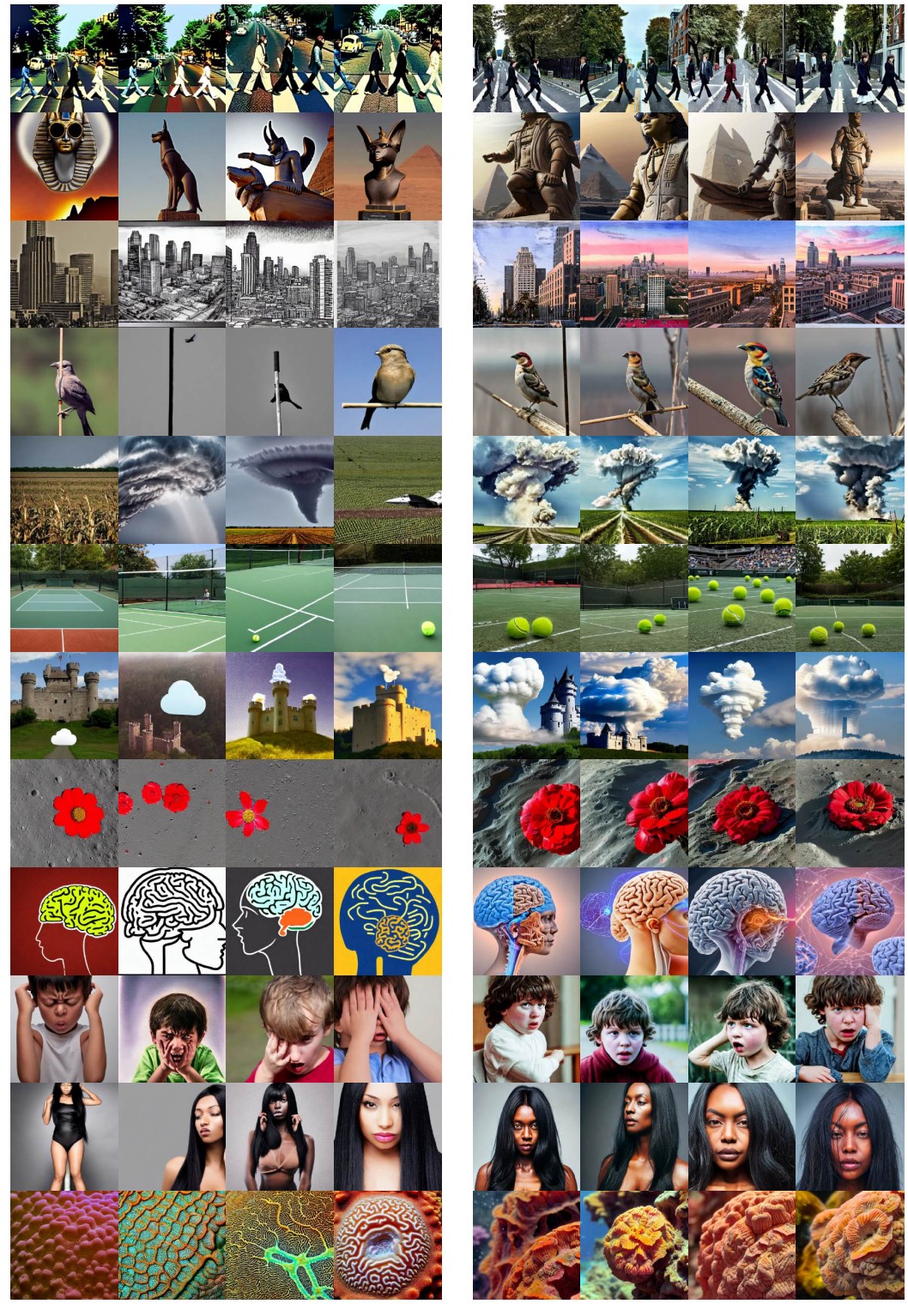

Figure 12: More examples of SD1.5 generations **before** and **after** tuning on Alchemist. Zoom in for the best view.

**Figure 13 prompts**

1. *"A blue Porsche 356 parked in front of a yellow brick wall"*
2. *"a flower with a cat's face in the middle"*
3. *"a flower with large yellow petals"*
4. *"A photo of an Athenian vase with a painting of pandas playing basketball in the style of Egyptian hieroglyphics."*
5. *"a teddy bear on a skateboard in times square"*
6. *"a red sports car on the road"*
7. *"an airplane flying into a cloud that looks like monster"*
8. *"graffiti of a funny dog on a street wall"*
9. *"a laptop screen showing a bunch of photographs"*
10. *"a view of the Kremlin on a sunny day"*
11. *"a lavender backpack with a triceratops stuffed animal head on top"*
12. *"Superman shaking hands with Spiderman"*

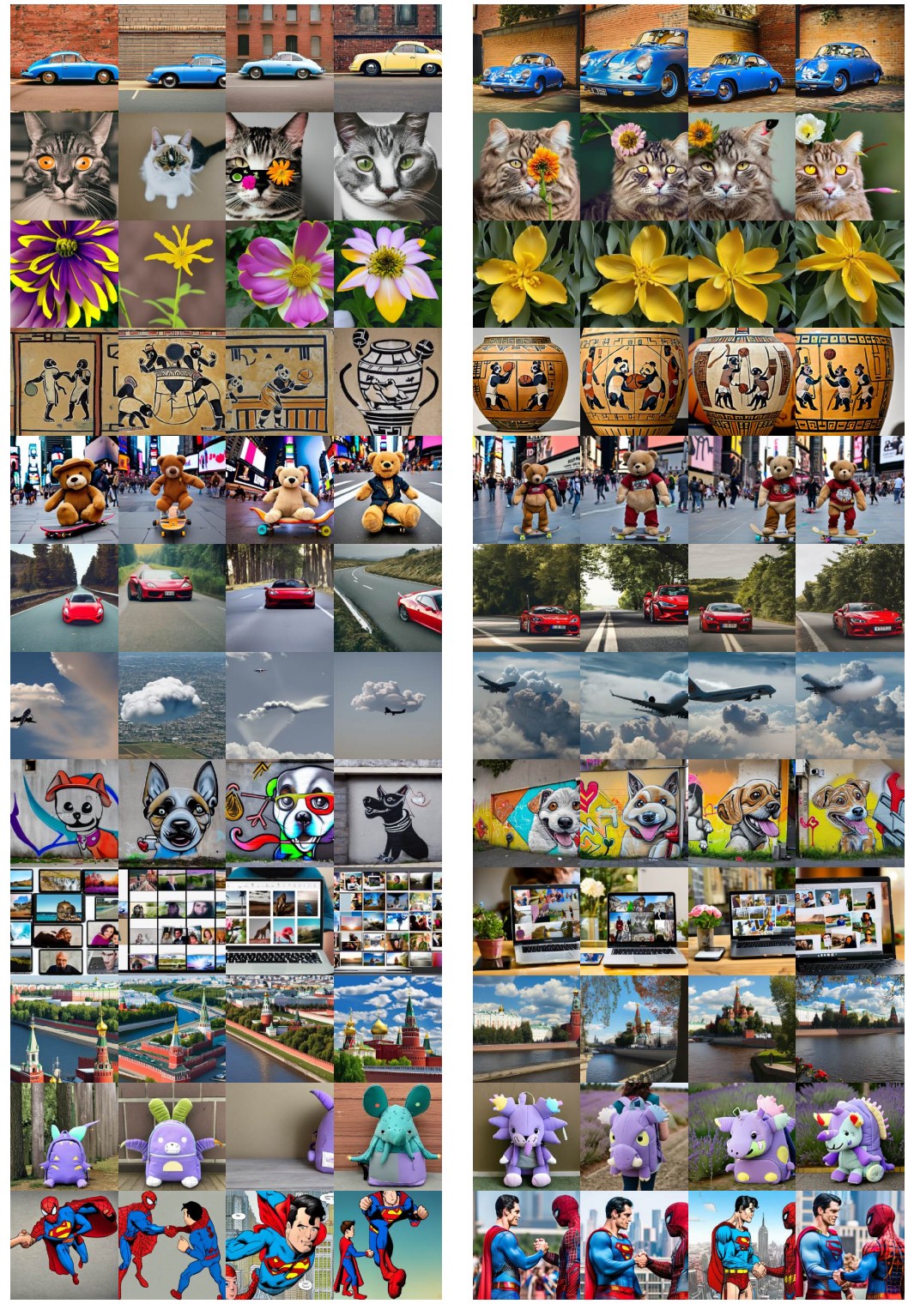

Figure 13: More examples of SD2.1 generations **before** and **after** tuning on Alchemist. Zoom in for the best view.

**Figure 14 prompts**

1. *"a chimpanzee wearing a bowtie and playing a piano"*
2. *"a white towel"*
3. *"a cat licking a large felt ball with a drawing of the Eiffel Tower on it"*
4. *"a man and a woman standing in the back up an old pickup truck"*
5. *"robots meditating"*
6. *"the silhouette of an elephant"*
7. *"A raccoon wearing formal clothes, wearing a top hat and holding a cane. The raccoon is holding a garbage bag. Oil painting in the style of Vincent Van Gogh."*
8. *"five red balls on a table"*
9. *"a pumpkin with a candle in it"*
10. *"A close-up of two mantis wearing karate uniforms and fighting, jumping over a waterfall."*
11. *"a yellow box to the right of a blue sphere"*
12. *"a futuristic city in synthwave style"*

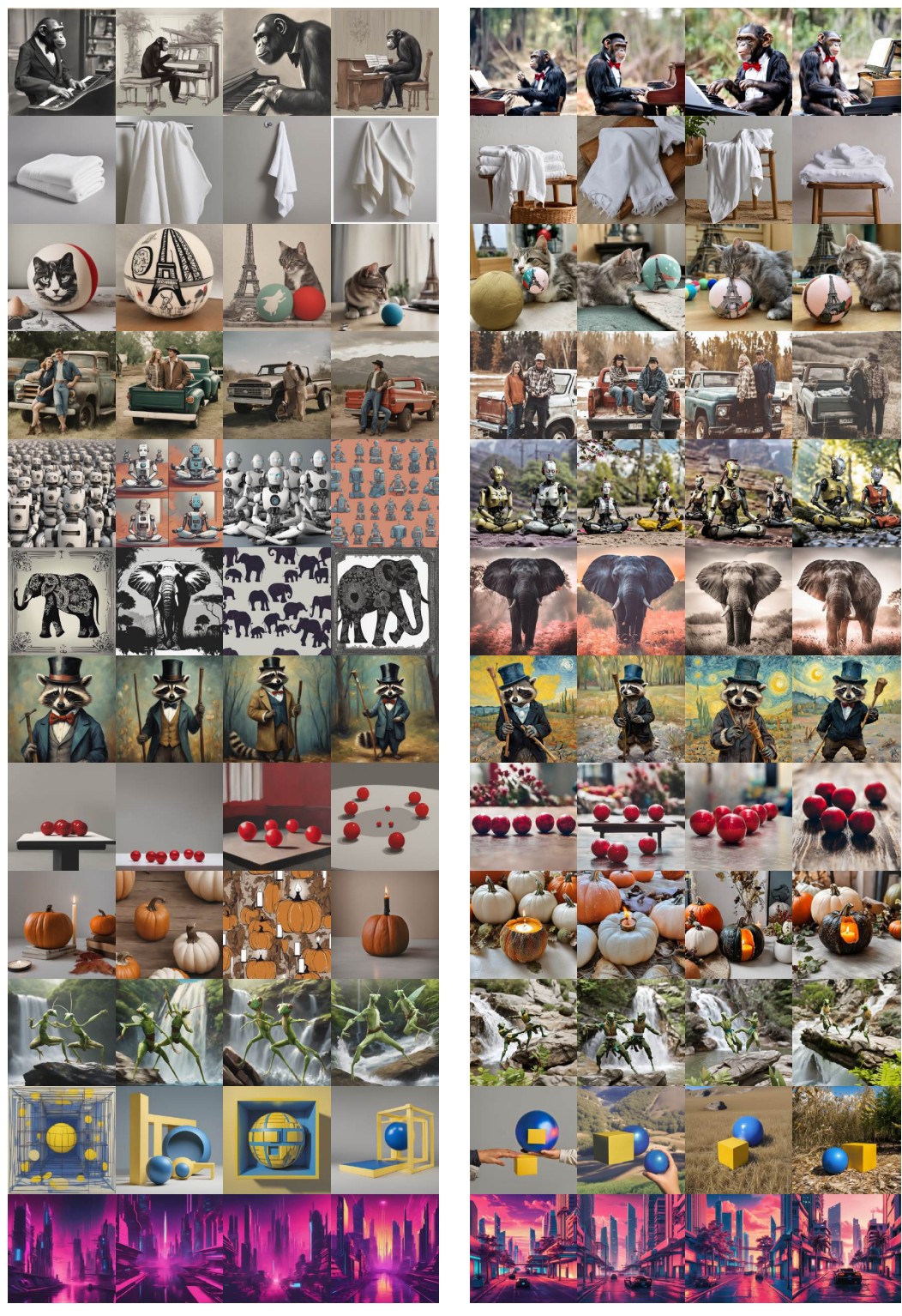

Figure 14: More examples of SDXL generations **before** and **after** tuning on Alchemist. Zoom in for the best view.

**Figure 15 prompts**

1. *"a witch riding a broom"*
2. *"A heart made of cookie"*
3. *"an airplane flying into a cloud that looks like monster"*
4. *"a peaceful lakeside landscape"*
5. *"a lavender backpack with a triceratops stuffed animal head on top"*
6. *"a red block to the left of a blue pyramid"*
7. *"a black dog sitting between a bush and a pair of green pants standing up with nobody inside them"*
8. *"a blue t-shirt"*
9. *"a woman with a dog puppet and a cat puppet"*
10. *"a yield sign"*
11. *"A photo of an astronaut riding a horse in the forest. There is a river in front of them with water lilies."*
12. *"a tiger standing by some flowers"*

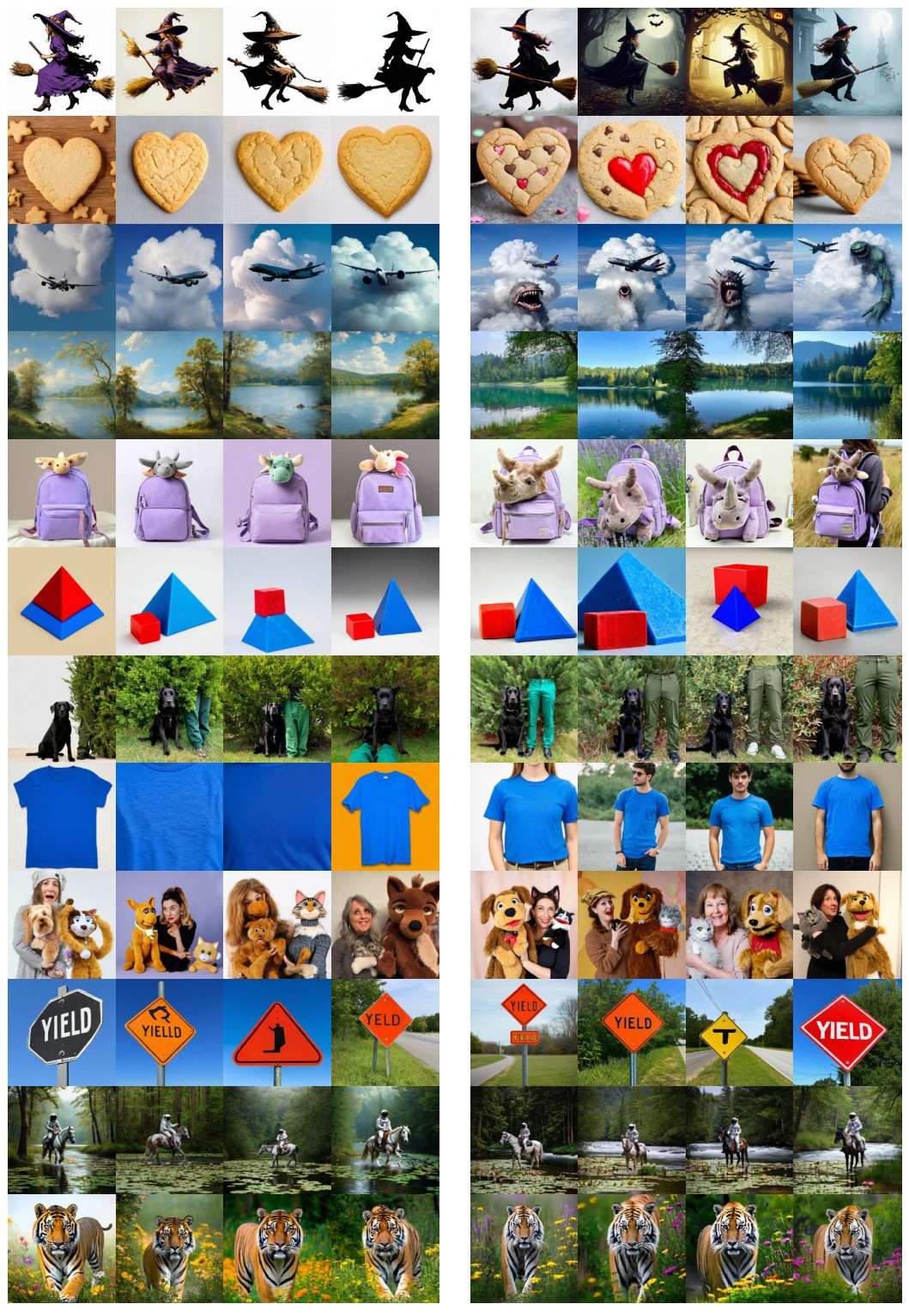

Figure 15: More examples of SD3.5 Medium generations **before** and **after** tuning on Alchemist. Zoom in for the best view.

**Figure 16 prompts**

1. *"The sunset on the beach is wonderful"*
2. *"a view of the Earth from the moon"*
3. *"A punk rock squirrel in a studded leather jacket shouting into a microphone while standing on a lily pad"*
4. *"Gandalf saying you shall not pass"*
5. *"a prop plane flying low over the Great Wall"*
6. *"a marine iguana crossing the street"*
7. *"a large white yacht tossed about in a stormy sea"*
8. *"the Parthenon in front of the Great Pyramid"*
9. *"a yellow wall"*
10. *"a chimpanzee wearing a bowtie and playing a piano"*
11. *"a tiny dragon landing on a knight's shield"*
12. *"A teddy bear wearing a motorcycle helmet and cape is standing in front of Loch Awe with Kilchurn Castle behind him"*

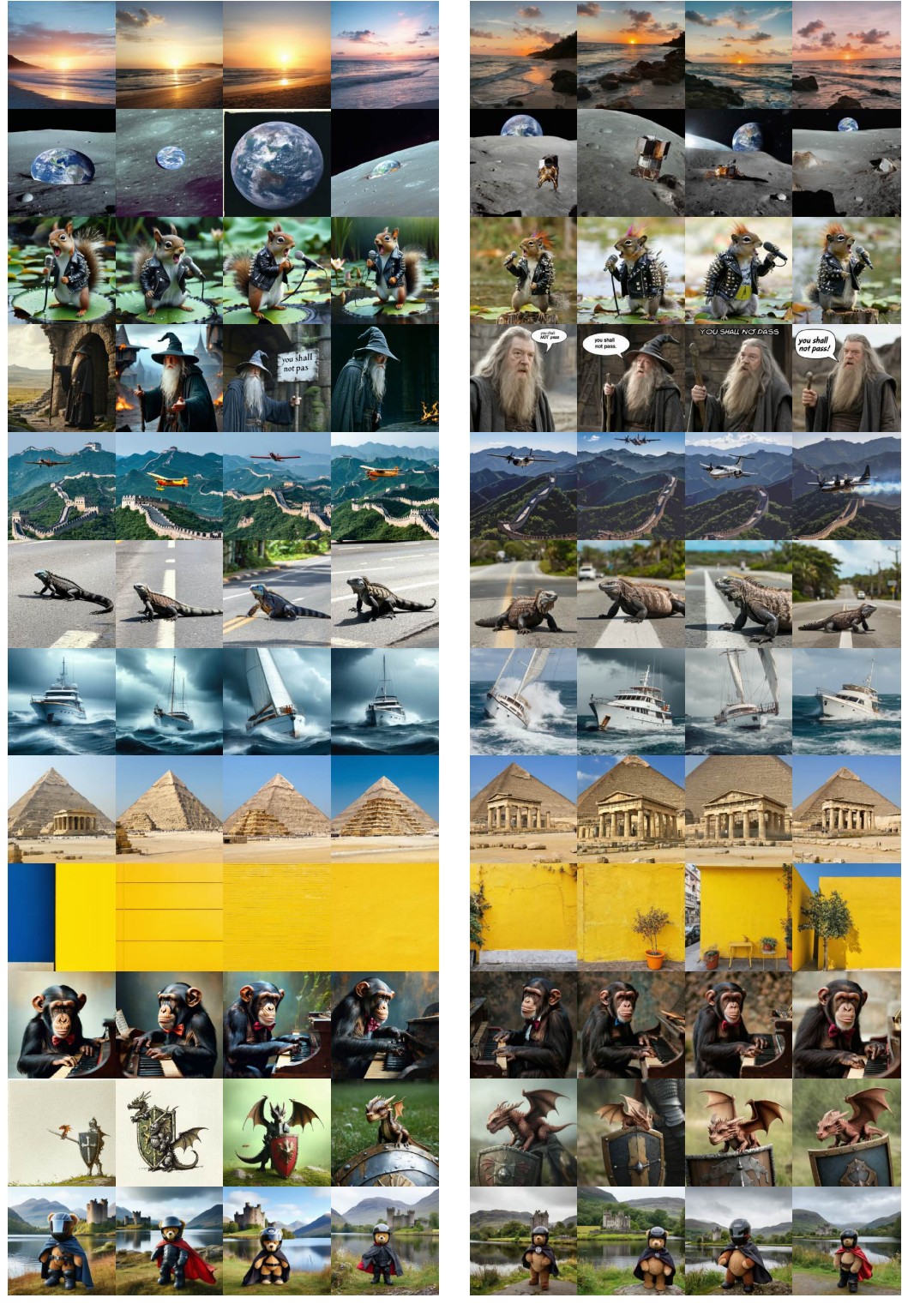

Figure 16: More examples of SD3.5 Large generations **before** and **after** tuning on Alchemist. Zoom in for the best view.

