# OpenReview forum: "Alchemist: Turning Public Text-to-Image Data into Generative Gold"
_NeurIPS.cc/2025/Datasets_and_Benchmarks_Track — NeurIPS 2025 Datasets and Benchmarks Track poster_

### Official Review · Reviewer_Sgy6 · 2025-06-28

**Ethics Flags:** Discrimination, bias, and fairness
**Rating:** 5
**Confidence:** 4

**Summary:**

This paper presents Alchemist, a compact yet highly effective supervised finetuning dataset created using a novel dataset curation methodology. The proposed method filters billions of images down to a compact dataset of 3,350 images through a multi-stage, image-based filtering pipeline. Finetuning five publicly available text-to-image models with the Alchemist dataset demonstrates performance improvements over alternative datasets.

**Dataset Code Accessibility:**

Partly

**Dataset Code Comments:**

The dataset is accessible via the provided link. However, the code for the curation methodology is not publicly available. Publishing the filtering code would significantly enhance transparency and reproducibility. Additionally, as noted in the weaknesses, several aspects of the curation methodology, such as prompt selection and calibration set availability, would benefit from improved documentation or release.

**Ethical Comments:**

The primary ethical concern relates to the limited number of human annotators (three) involved in the evaluation. If the paper provides a thorough explanation and justification, this concern can be mitigated. However, as it stands, the reviewer is still concerned about the fairness and robustness of the evaluation process.

**Ethical Considerations:**

Yes, there are ethics concerns that require attention by the authors

**Final Justification:**

After the rebuttal, all of my concerns, including the initial ethical concerns, have been resolved. Specifically, my issues regarding the dataset curation methodology, such as prompt selection, the calibration set, and the caption generation model, have been fully addressed. Additionally, my concerns about the human evaluation and the use of the visual language model for evaluation have also been resolved.

I prefer to keep my scores unchanged, as I believe the paper’s contributions remain substantial.

**Limitations Weaknesses:**

Two main areas of concern arise from the dataset curation methodology and the human evaluation process:
- Issues with the Curation Methodology
Several specific concerns are outlined below:
1. Prompt Selection (L.142): The curation process uses multi-keyword prompts to evoke target visual qualities. However, the supplementary material (Section B.2) shows that the number of prompts is limited. This raises questions about possible biases introduced by the prompt choices. Could the authors ablate or expand the set of prompts? Is there a way to quantitatively assess their impact?
2. Calibration Set (L.145): A calibration set of 1,000 images is used to distinguish “higher-quality” and “lower-quality” images. Releasing this set would enhance reproducibility and transparency. Can the authors consider publishing it?
3. Caption Style (L.157): A preliminary study suggests that captions resembling user prompts yield better results. This is an interesting finding, but it is only briefly mentioned. More elaboration, either in the main paper or supplementary material, would help clarify its implications.
4. Caption Generation Model (L.160): The dataset’s captions are generated using a proprietary model. Could the authors evaluate or provide alternatives using publicly available captioning models? Understanding their impact on dataset quality would improve reproducibility and provide valuable insights.

- Issues in evaluation:
1. Human Evaluation Sample Size: The human evaluation was conducted with only three expert annotators, which seems too small to ensure fairness or robustness. Are three evaluators sufficient? If so, the authors should provide justification and details. If not, alternatives or additional validation are needed.
2. Use of Visual Language Models: Could large vision-language models be used to perform side-by-side evaluations? Measuring the correlation between such models and human evaluations could strengthen the evaluation pipeline and enhance the paper’s contribution.

**Strengths Contributions:**

- Clear Contributions Supported by Experiments:
The paper clearly describes both the dataset curation methodology and the resulting Alchemist dataset. The proposed filtering approach is novel in that it relies entirely on image-based criteria. Finetuning with Alchemist consistently outperforms baselines in both human evaluations and automated metrics. While some evaluation metrics show a decline, the paper provides reasonable explanations for these cases in the discussion section.

- Novel Dataset Curation Methodology:
The dataset is curated through a multi-stage image-based filtering pipeline. While multi-stage filtering itself may not be entirely novel, the use of a diffusion-based quality estimator in the final stage introduces a unique aspect. This innovative filtering approach demonstrates its value by enhancing finetuning performance.

- Extensive Experiments and Sound Discussion:
The experimental results are presented clearly and include both human evaluations and automated metrics. The experiments effectively validate the benefits of supervised finetuning using Alchemist. Furthermore, the paper openly discusses areas where performance declines after SFT and provides well-reasoned explanations.

---

> ### Author Rebuttal · Authors · 2025-07-31
>
> We are grateful to the reviewer for the meticulous and considerate feedback. The concerns are addressed below.
>
> **Weaknesses**:
>
> - **Curation Methodology**
>     1. **On prompt selection**. We thank the reviewer for raising this critical point about prompt selection bias. This is an important consideration, and we designed our methodology with this challenge in mind.
>
>         Firstly, to mitigate bias from any single concept, we intentionally designed our diffusion-based estimator prompt to be long and diverse, not narrow. As detailed in Appendix B.2, it incorporates a broad range of keywords associated with general visual quality ("high quality," "aesthetic," "complex," "photorealistic," etc.), rather than focusing on a specific, subjective style. The goal of this prompt is to activate a general-purpose "quality vector" within the diffusion model, not to steer the selection towards a niche aesthetic.
>
>         Secondly, while a direct ablation on multiple scoring prompts is computationally prohibitive as it would require re-scoring all 300 million candidate images and re-running all fine-tuning experiments for each prompt variation. We use our comprehensive downstream evaluations to indirectly validate our prompt choice. The results demonstrate that Alchemist-tuned models did not collapse or overfit:
>
>         - **Qualitative analysis** (Figure 3, Appendix H) shows preserved stylistic and content diversity.
>
>         - **Human SbS evaluations** and **CLIP Scores** confirmed that **Image-Text Relevance** was not degraded.
>
>         - **FID scores** remained stable, suggesting no significant distributional shift away from the baseline.
>
>         The combination of these results strongly indicates that our chosen prompt successfully guided the selection towards broadly applicable quality without introducing a narrow, restrictive bias. While we did not perform a systematic ablation, we can confirm that this prompt was selected after several preliminary iterations showed its effectiveness in capturing general quality. We will add a sentence to the manuscript to clarify our rationale for the prompt design and how its effectiveness is validated by the downstream results.
>
>     2. **On the calibration set**. We thank the reviewer for this excellent suggestion. We fully agree that releasing the 1,000-image calibration set would be highly beneficial for reproducibility and for future research building on our methodology.
>
>        We are fully committed to making this set public. However, as the images are derived from web-scraped data, they are subject to a final legal and ethical review before they can be redistributed, a process we were unable to complete within the rebuttal period.
>
>        We have already initiated this review. Upon its successful completion, we will release the full calibration set, including image identifiers and our labels, to the same Hugging Face repository that hosts the Alchemist dataset and our fine-tuned models. We will also add a note to the manuscript stating our intention to release this data pending this final review.
>
>   3. **On captions style and caption generation model**. To evaluate how captioning models affect our dataset, we recaptioned the Alchemist dataset using two open-source state-of-the-art models: Qwen2.5VL 72B and InternVL2 26B.
>
>       We then measured the impact by comparing SDXL and SD3.5 Medium models fine-tuned on the recaptioned data. As in the paper, we conducted SbS comparison on the PartiPrompts test set and color-coded statistically significant changes (p-value<0.05) with (green) and (red), while (gray) means no statistically significant changes.
>
>         **Side-by-side comparison**:
>
>         | Model| Rel. | Aes. | Comp. | Fidel. |
>         |------------------------------|------|------|-------|--------|
>         |  SDXL-Alchemist-QwenVL_caps     |  | | |  |
>         | vs SDXL-original             | 0.55 (green)  | 0.57 (green) | 0.71 (green) | 0.52 (gray)  |
>         | vs SDXL-Alchemist            | 0.52 (gray) | 0.44 (red) | 0.40 (red) | 0.57 (green)  |
>         |------------------------------|------|------|-------|--------|
>         | SDXL-Alchemist-InternVL_caps       | | | |  |
>         | vs SDXL-original     | 0.53 (gray)  | 0.56 (green) | 0.71 (green) | 0.51 (gray)  |
>         | vs SDXL-Alchemist            | 0.51 (gray) | 0.47 (gray) | 0.41 (red) | 0.52 (gray) |
>
>         | Model  | Rel. | Aes. | Comp. | Fidel. |
>         |------------------------------|------|------|-------|--------|
>         |  SD3.5M-Alchemist-QwenVL_caps         |  | | |  |
>         | vs SD3.5M-original    | 0.54 (gray)  | 0.61 (green) | 0.63 (green) | 0.50 (gray)  |
>         | vs SD3.5M-Alchemist     | 0.51 (gray) | 0.48 (gray) | 0.44 (red) | 0.46 (gray)  |
>         |------------------------------|------|------|-------|--------|
>         |  SD3.5M-Alchemist-InternVL_caps         |  | | |  |
>         | vs SD3.5M-original    | 0.52 (gray)  | 0.64 (green) | 0.69 (green) | 0.44 (red)  |
>         | vs SD3.5M-Alchemist     | 0.53 (gray) | 0.49 (gray) | 0.47 (gray) | 0.43 (red)  |
>
>         While the new models don't quite match the aesthetic and complexity levels of the original Alchemist-tuned versions (despite occasionally producing fewer artifacts), using captions from open-source models still leads to significant quality improvements in generation.
>
>
> - **Issues in evaluation**
>
>     1. **On human evaluation sample size**. Please note that three different annotators evaluate a **single** image pair, not all pairs. Our evaluations are conducted by a pool of **more than 1000** expert annotators, with each image pair assessed by a randomly selected triplet (note that assigning three annotators per pair is a well-established practice in the literature, e.g., [1, 2]). Thus, we believe our procedure ensures a fair and robust human preference study. We will clarify these details in the revision.
>
>   2. **On use of Visual Language Models**. The use of large vision-language models (VLMs) as proxies for human evaluation is indeed a rapidly evolving and important area of research. To investigate this, we conducted an additional automated evaluation during the rebuttal period.
>
>      We adopted the VLM-as-a-judge methodology and computed a text-to-image variant of VIEScore [3], using GPT-4o as the backbone due to its demonstrated strong correlation with human judgments. VIEScore assesses two primary dimensions: Semantic Consistency (SC), which aligns with our **Relevance** criterion, and Perceptual Quality (PQ), which most closely corresponds to our **Fidelity** criterion (absence of defects).
>
>      For each model pair and prompt, we used GPT-4o to determine a winner, loser, or tie based on these two dimensions. The aggregated results, analyzed with the same statistical methodology as our human evaluations, are presented below:
>
>      | SD1.5-Alchemist | SC | PQ|
>      |---------------------|----------------------|--------------------|
>      | vs SD1.5-original  | 0.52 (gray)                  | 0.60 (green)                |
>
>      | SD2.1-Alchemist | SC| PQ|
>      |---------------------|----------------------|--------------------|
>      | vs SD2.1-original  | 0.52 (gray)                 |  0.62 (green)               |
>
>      | SDXL-Alchemist | SC| PQ|
>      |---------------------|----------------------|--------------------|
>      | vs SDXL-original  | 0.5 (gray)                | 0.5 (gray)                |
>
>      | SD3.5M-Alchemist | SQ| PQ|
>      |---------------------|----------------------|--------------------|
>      | vs SD3.5M-original  | 0.52 (gray)                 | 0.52 (gray)               |
>
>      | SD3.5L-Alchemist | SC | PQ|
>      |---------------------|----------------------|--------------------|
>      | vs SD3.5L-original  | 0.5 (gray)                  | 0.52 (gray)                |
>
>      The results from this VLM-based evaluation are broadly consistent with our human assessments for the corresponding criteria (Relevance and Fidelity). The VLM-judge confirms the preservation of textual relevance and, interestingly, assesses our model's fidelity more favorably than our expert human annotators.
>
>      This provides strong evidence that modern VLMs can serve as a reliable and scalable proxy for evaluating objective aspects of T2I model performance. However, we also note that current VLM evaluation frameworks, including VIEScore, are primarily designed to assess more objective criteria like prompt alignment and artifact detection. Reliably capturing more subjective and nuanced human preferences, such as **Aesthetic Appeal** and **Image Complexity**, remains a significant challenge and an important direction for future research in the field [3, 4].
>
>       We will add a new subsection to the paper detailing this VLM-based evaluation, which strengthens our findings by corroborating them with a state-of-the-art automated method.
>
> **References**:
>
> [1] Han, Jian, et al. "Infinity: Scaling Bitwise AutoRegressive Modeling for High-Resolution Image Synthesis". IEEE/CVF Conference on Computer Vision and Pattern Recognition (2025): 15733-15744.
>
> [2] Shi, Zhan, et al. "Improving image captioning with better use of captions." arXiv preprint arXiv:2006.11807 (2020).
>
> [3] Ku, Max, et al. "VIEScore: Towards Explainable Metrics for Conditional Image Synthesis Evaluation." 62nd Annual Meeting of the Association for Computational Linguistics (Volume 1: Long Papers) (2024): 12268–12290.
>
> [4] Lin, Zhiqiu, et al. "Evaluating text-to-visual generation with image-to-text generation." European Conference on Computer Vision. Cham: Springer Nature Switzerland (2024).

---

> > ### Comment · Reviewer_Sgy6 · 2025-08-02
> >
> > Thank you for addressing my concerns regarding the dataset curation methodology and human evaluation feedback. I appreciate the additional experiments and clarifications, which have helped resolve my misunderstanding. I no longer have any ethical concerns.
> >
> > Please ensure that the paper and supplementary material are revised to reflect the updates on prompt selection, the caption generation model, the calibration set, and the use of the visual language model.

---

### Official Review · Reviewer_KySp · 2025-07-02

**Rating:** 4
**Confidence:** 3

**Summary:**

The paper proposes leveraging a pre-trained generative model to identify high-impact training samples for constructing a high-quality and general-purpose text-to-image SFT dataset, Alchemist (comprising 3,350 samples). The authors select this subset by filtering images that demonstrate high cross-attention activations associated with prompts describing specific target characteristics, such as "high quality" and "aesthetic." They conduct experiments to extract 3,350 target samples from a pool of 300 million images for SFT.

**Dataset Code Accessibility:**

NA; not applicable to this submission (e.g., no new dataset, benchmark, code, or data provided)

**Ethical Considerations:**

No, there are no or only very minor ethics concerns

**Final Justification:**

The authors have provided a thorough rebuttal that addresses my earlier concerns. I have raised my rating to borderline accept. I recommend incorporating these responses into the final version, including: (1) the prompts used for each visual sample, (2) an analysis of the high-quality SFT samples, and (3) the additional experimental results.

**Limitations Weaknesses:**

1. It is suggested to include the prompt in all figures to show the model's prompt adherence.
2. The authors mention subtle specific characteristics of good SFT samples in Lines 30–32. Including concrete visual examples would help to better demonstrate these characteristics.
3. Based on the visual samples shown in Figure 3 and Figures 7–9, the diversity of generated samples appears to decrease after SFT. The authors should include quantitative evaluation and discussion regarding diversity.
4. According to the quantitative results reported in Table 1, the improvements of Alchemist-tuned models are minor compared with LAION-tuned models.

**Strengths Contributions:**

1. This paper addresses the important task of filtering high-priority training samples for SFT.
2. The authors conduct extensive experiments across multiple baselines, including SD1.5, SD2.1, SDXL, SD3.5M, and SD3.5L.

---

> ### Author Rebuttal · Authors · 2025-07-31
>
> Thank you for the thorough and insightful review. We have addressed your concerns as follows.
>
> **Weaknesses**:
>
> 1. **On prompts in all figures**. We have reviewed the figures and agree that including the prompt for the illustrative examples in Figure 1 would be beneficial. For all other figures presenting qualitative results (e.g., Figure 3, Appendix H), the corresponding prompts are already provided.
>
>     We will update the caption of Figure 1 in the revised manuscript to include the prompts used for those specific images.
>
>     At the meantime, prompts for the Figure 1 are the following:
>     - “a subway train coming out of a tunnel”
>     - “a man standing under a tree”
>     - “a girl with long curly blonde hair and sunglasses”
>     - “energy in hand-blown glass vessels, central potion bottle with 'ALCHEMIST' etched directly into glass, ambient magical particles, dramatic backlighting casting colored shadows, exquisite detail, professional product photography style with fantasy elements”
>
> 2. **On characteristics of good SFT samples**. We thank the reviewer for this excellent point. The difficulty in explicitly defining and verbalizing the "specific characteristics" of good SFT samples is precisely the core motivation for our diffusion-based approach. If these traits were easily identifiable through simple heuristics, a sophisticated estimator would be unnecessary.
>
>     However, we can describe the high-level patterns our method aims to capture. Our preliminary analyses revealed that top-performing SFT samples often possess a complex blend of attributes, such as:
>
>     - **High Technical Quality**: Sharp focus, good lighting, and absence of compression artifacts.
>     - **Aesthetic Coherence**: Harmonious color palettes and strong compositional structure.
>     - **Semantic Richness**: A moderate level of detail and complexity without being visually chaotic.
>
>     While any single attribute can be measured by existing tools, it is the holistic combination of these traits that our method seeks. Our diffusion-based estimator, prompted with a diverse set of keywords related to these patterns (as detailed in Appendix B), learns to identify images that embody this subtle, holistic quality.
>
>     To make this clearer in the manuscript, we will add a figure in the appendix presenting a few examples of images that received high scores from our estimator alongside those that scored low, with brief annotations highlighting these distinguishing characteristics. This will provide a qualitative illustration of the nuanced features our method successfully identifies.
>
>  3. **On diversity of generated images**. To quantitatively investigate this, we conducted an analysis of intra-prompt diversity, following the methodology in [1]. For each prompt in our test set, we generated N=4 images using different seeds. We then computed the average pairwise cosine distance between the feature embeddings of these images (extracted using a CLIP ViT-L model). A higher score indicates greater diversity. The results are presented below.
>
>     |        | original    | Alchemist-tuned |
>     |--------|-------------|-----------------|
>     | SD1.5  | 0.37, [0.36; 0.39] |   0.26, [0.25; 0.27]  |
>     | SD2.1  | 0.34, [0.33; 0.35] |  0.20, [0.19; 0.21]   |
>     | SDXL   | 0.26, [0.25; 0.27 ] |  0.22, [0.21; 0.23] |
>     | SD3.5M | 0.22, [0.21; 0.23]  |  0.18, [0.17; 0.19]  |
>     | SD3.5L | 0.20, [0.19, 0.21]  | 0.17, [0.16, 0.18]  |
>
>     The results confirm a noticeable decrease in the diversity metric for Alchemist-tuned models. However, we posit that this reduction does not signify a loss of global stylistic or conceptual coverage. We believe, to some extent, these results reflect the model's increased reliability and its convergence towards high-quality outputs. Baseline models often exhibit higher "error diversity" by producing off-prompt, lower-quality, or nonsensical generations, which, while distant in feature space, do not represent a desirable creative range. Alchemist fine-tuning reduces this undesirable diversity by consistently generating high-fidelity images that are more thematically coherent with the prompt.
>
>     This interpretation is strongly supported by our other findings. A true collapse in diversity (i.e., mode collapse) would lead to a sharp decline in Image-Text Relevance and CLIP Scores, as the model would fail to generate a wide range of concepts accurately. As shown in our main results (Table 1), these metrics remained stable and robust after fine-tuning. This indicates that the model's ability to address diverse prompts is fully preserved, and the measured decrease in diversity is partially attributable to the elimination of low-quality failure modes.
>
>     We will add a new subsection with this quantitative diversity analysis and discussion to our paper to clarify this important nuance.
>
> 4. **On the magnitude of improvement from tuning with Alchemist**. We thank the reviewer for this careful reading of our quantitative results. We agree that when viewed through the lens of certain automated metrics in Table 1, the performance gains of Alchemist-tuned models over LAION-tuned models can appear modest. However, we argue that this highlights a well-known limitation of current automated metrics rather than a lack of significant improvement.
> Our primary evaluation relies on a rigorous, quantitative human side-by-side (SbS) comparison precisely because automated metrics often fail to capture nuanced perceptual qualities. **In these human evaluations, the improvements are substantial and statistically significant.** As shown in Table 1, Alchemist-tuned models consistently outperformed LAION-tuned models in both **Aesthetic Quality** and **Image Complexity** by significant margins (win rate advantages of +12% to +20%). This demonstrates a clear and meaningful perceptual enhancement that automated metrics are not fully sensitive to.
> The observed discrepancy with automated metrics is not entirely unexpected [2]. Metrics like FID measure distributional feature similarity and are not designed to assess the aesthetic quality or complexity of individual images. Furthermore, learned preference models like ImageReward and HPSv2, while valuable, are known to have their own inherent biases stemming from their training data and annotation procedures, as acknowledged by their authors [3, 4].
>
>     Therefore, while we report automated metrics for completeness, we believe the strong and consistent signal from our multi-aspect human evaluations provides a more accurate and reliable assessment of Alchemist's significant positive impact. We will add a sentence to the "Results" section to more clearly contextualize the automated metrics in light of the human evaluation findings.
>
> **References**:
>
> [1] Boutin, Victor, et al. "Diversity vs. Recognizability: Human-like generalization in one-shot generative models." Advances in Neural Information Processing Systems 35 (2022): 20933-20946.
>
> [2] Borji, Ali. "Pros and cons of GAN evaluation measures." Computer vision and image understanding 179 (2019): 41-65.
>
> [3] Xu, Jiazheng, et al. "Imagereward: Learning and evaluating human preferences for text-to-image generation." Advances in Neural Information Processing Systems 36 (2023): 15903-15935.
>
> [4] Wu, Xiaoshi, et al. "Human preference score v2: A solid benchmark for evaluating human preferences of text-to-image synthesis." arXiv preprint arXiv:2306.09341 (2023).

---

> ### Comment · Area_Chair_dFAp · 2025-08-09
>
> Dear Reviewer KySp,
>
> The reviewer-author discussion will end in 24 hours. Please respond to authors' rebuttal and participate in the discussion before it closes.
>
> Best,
>
> AC

---

### Official Review · Reviewer_SHi1 · 2025-07-02

**Rating:** 4
**Confidence:** 4

**Summary:**

This paper introduced a method that uses a pre-trained diffusion model to identify high-impact training samples, addressing the scarcity of effective public datasets for SFT of text-to-image models. The authors proposed the Alchemist dataset with 3350 samples by filtering web-scale data through safety checks, deduplication, and a diffusion-guided scoring system based on cross-attention activations. For evaluation, the authors finetuned five Stable Diffusion variants with Alchemist, significantly improved aesthetic quality and image complexity without sacrificing diversity, outperforming size-matched alternatives like LAION-Aesthetics despite minor fidelity trade-offs in newer architectures.

**Additional Feedback:**

1. **Dataset curation methodology**

   The diffusion-based scoring function relies on manually selected keywords and a small calibration set. How were these keywords chosen, and what validation was done to ensure they accurately reflect "high-impact" samples for SFT? Could the manual selection introduce bias, and have the authors considered automated keyword selection methods?

2. **Dataset size ablation**

   The ablation study shows that larger Alchemist variants perform worse, but what about even smaller subsets? Could there be an ablation study for the sweet spot below 3,350 samples that still delivers significant improvements?

3. **Generalizability and model diversity**

   The experiments exclusively use Stable Diffusion variants. Have you considered testing Alchemist on other text-to-image architectures?

   The test prompts are sourced from PartiPrompts. While this is a standardized benchmark, have you evaluated the model on other datasets (e.g., MS-COCO, DrawBench) to assess performance across a broader range of real-world scenarios?

**Dataset Code Accessibility:**

Yes

**Ethical Comments:**

No significant ethical concerns are identified.

**Ethical Considerations:**

No, there are no or only very minor ethics concerns

**Limitations Weaknesses:**

1. **Dataset size ablation study**

   The dataset size ablation study shows that larger variants of Alchemist performed worse(line 249), but the paper doesn't explore if even smaller datasets could be as effective.

2. **Subjectivity in Quality Estimation**

   The paper uses TOPIQ(line 130) and a scoring function based on cross-attention activations(line 141). However, the scoring relies on manually defined keywords (e.g., “high quality,” “artistic”) and a small calibration set (1,000 images). The subjectivity in defining “high-impact” samples isn't thoroughly addressed. This may introduce bias and may hinder generalizability.

3. **Limited prompt diversity in evaluation**

   The test prompts are from PartiPrompts(line 189). The exclusive reliance on PartiPrompts for test prompts may raise concerns about generalizability, as this dataset might not cover all real-world scenarios, thereby affecting the validity of the results.

   While PartiPrompts is a standardized benchmark, incorporating other T2I datasets (e.g., MS-COCO, DrawBench) would be beneficial for a more comprehensive evaluation.

4. **Limited Model Diversity**

   The base models are all Stable Diffusion variants. Testing on other architectures remains untested, limiting claims of "general-purpose" applicability.

**Strengths Contributions:**

1. **Key strengths and contributions**

   This paper's primary strength is its innovative methodology leveraging pre-trained diffusion models as quality estimators. The work uses cross-attention activations to curate Alchemist, a compact open SFT dataset that demonstrably elevates aesthetics and complexity across multiple models, proving that sample quality trumps dataset scale.

2. **Relationship to existing benchmarks**

   Alchemist fills a critical gap as a compact general-purpose public SFT dataset. It outperforms size-matched benchmarks like LAION-Aesthetics and avoids the overfitting of domain-specific datasets (e.g., Danbooru), while rivaling proprietary models (e.g., FLUX) with fewer parameters.

3. **Presentation**

   The presentation is rigorously transparent. The paper methodically detailed the multi-stage filtering pipeline and validated results through statistically robust human evaluations and automated metrics. The authors candidly discussed limitations, ensured reproducibility via open-sourced data, and documented compute and ethics compliance.

---

> ### Author Rebuttal · Authors · 2025-07-31
>
> We thank the reviewer for a detailed and thoughtful review. We address the concerns below:
>
> **Weaknesses**:
>
> 1. **Dataset size ablation study**. To investigate this, we conducted an additional ablation study on smaller dataset sizes during the rebuttal period. We created three smaller variants of Alchemist by taking the top 500, 1,000, and 1,500 samples as ranked by our diffusion-based estimator. We then fine-tuned two representative models, SDXL and SD3.5 Medium, on these smaller datasets. For these runs, we linearly scaled down the number of training steps while keeping all other hyperparameters consistent.
>
>     We compared the models tuned on these smaller datasets against the model tuned on our original 3,350-sample Alchemist (Alchemist-3k). The results of our side-by-side human evaluation are presented below.
>
>     **Side-by-side comparison**:
>     | Model| Rel. | Aes. | Comp. | Fidel. |
>     |-|-|-|-|-|
>     | SDXL-Alchemist-1.5k|  | | |  |
>     | vs SDXL-original  | 0.53 (gray)  | 0.60 (green) | 0.80 (green) | 0.53 (gray)  |
>     | vs SDXL-Alchemist-3k | 0.50 (gray) | 0.49 (gray) | 0.46 (gray) | 0.57 (green)  |
>     |------------------------------|------|------|-------|--------|
>     | SDXL-Alchemist-1k            |  | | |  |
>     | vs SDXL-original             | 0.52 (gray)  | 0.57 (green) | 0.72 (green) | 0.52 (gray)  |
>     | vs SDXL-Alchemist-3k           | 0.52 (gray) | 0.45 (red) | 0.41 (red) | 0.59 (green)  |
>     |------------------------------|------|------|-------|--------|
>     | SDXL-Alchemist-500            |  | | |  |
>     | vs SDXL-original             | 0.54 (gray)  | 0.63 (gray) | 0.69 (green) | 0.54 (gray)  |
>     | vs SDXL-Alchemist-3k           | 0.52 (gray) | 0.41 (red) | 0.36 (red) | 0.60 (green)  |
>
>     | Model| Rel. | Aes. | Comp. | Fidel. |
>     |-|-|-|-|-|
>     | vs SD3.5M-Alchemist-1.5k | | | |  |
>     | vs SD3.5M-original| 0.51 (gray)  | 0.68 (green) | 0.74 (green) | 0.47 (gray)  |
>     | vs SD3.5M-Alchemist-3k | 0.51 (gray) | 0.58 (green) | 0.57 (green) | 0.45 (gray) |
>     |------------------------------|------|------|-------|--------|
>     | vs SD3.5M-Alchemist-1k            | | | |  |
>     | vs SD3.5M-original             | 0.51 (gray) | 0.70 (green) | 0.77 (green)| 0.48 (gray) |
>     | vs SD3.5M-Alchemist-3k            | 0.5 (gray) | 0.58 (green) | 0.59 (green) | 0.49 (gray) |
>     |------------------------------|------|------|-------|--------|
>     | vs SD3.5M-Alchemist-500  | | | |  |
>     | vs SD3.5M-original | 0.52 (gray)  | 0.70 (green)| 0.79 (green) | 0.50 (gray) |
>     | vs SD3.5M-Alchemist-3k | 0.52 (gray) | 0.56 (green) | 0.54 (gray) | 0.49 (gray) |
>
>     While the 500-sample and 1,000-sample variants still show significant improvements over the baseline models, their performance relative to the 3,350-sample set is inconsistent. For instance, while the SD3.5 Medium model fine-tuned on 1k samples shows a competitive or even slightly improved performance, the SDXL model exhibits a clear trade-off, with gains in one aspect (e.g., Fidelity) coming at the cost of others (e.g., Aesthetics, Complexity).
>
>     This inconsistency across different model architectures suggests that while very small, highly curated datasets can be potent, they may not offer the same level of robust, general-purpose improvement. A larger set like our 3,350-sample Alchemist appears to provide a more stable and well-rounded enhancement across diverse models.
>
>     Furthermore, we hypothesize that fine-tuning on extremely small datasets, while potentially effective on some primary metrics, may carry a higher risk of negatively impacting other, unmeasured qualities, such as a more significant drop in generative diversity or overfitting to the few concepts present in the small set.
>
>     We will add a summary of this lower-bound ablation to our appendix to provide a complete picture of the impact of dataset size.
>
> 2. **On manually defined keywords and small calibration set**. We acknowledge the reviewer's point regarding the potential for bias from our prompt-based scoring function. This is a critical aspect of our methodology, and we made several deliberate design choices to mitigate this risk while still creating a powerful quality estimator.
>
>     - **Broad and Generic Prompt Design**: Rather than using a narrow, highly specific prompt, we intentionally constructed a long and diverse one. It incorporates a wide range of keywords associated with general visual quality ("high quality," "aesthetic," "complex," "photorealistic," etc.). This design aims to capture a broad consensus of positive attributes rather than overfitting to a single, subjective stylistic preference. The goal was to build an estimator for general technical and aesthetic excellence, not for a specific "look."
>
>     - **Purpose of the Calibration Set**: The 1,000-image calibration set is not used to train a new model but to identify the most discriminative, pre-existing features within the foundational diffusion model. By using a relatively small and randomly sampled set, we reduce the risk of discovering spurious correlations that might exist in a larger, more homogenous calibration set. This step is about feature selection from a vast, pre-learned space, not about learning the features themselves from our small set.
>
>     - **Validation Through Results**: Ultimately, the most effective validation of this approach is in its downstream performance. The fact that Alchemist fine-tuning preserves concept and style diversity and enhances a wide range of models (SD1.5 through SD3.5) without collapsing them into a single style suggests that our method successfully avoided introducing strong, restrictive biases.
>
>     We will amend the manuscript to more clearly articulate these design choices and their rationale for mitigating potential biases.
>
> 3. **On limited prompt diversity in evaluation**. We agree that evaluating on diverse prompt distributions is crucial for validating the general-purpose nature of our SFT dataset. To address this, we conducted an additional evaluation during the rebuttal period using the challenging DrawBench benchmark [1]. We performed the same side-by-side (SbS) human evaluation, comparing our Alchemist-tuned models against their baselines on the DrawBench prompt set. The results are summarized below:
>
>    **Side-by-side comparison**:
>
>    | **SD1.5-Alchemist** | Rel. | Aes. | Comp. | Fidel. |
>    |-|-|-|-|-|
>    |vs SD1.5-original|0.52 (gray)|0.66 (green)|0.77 (green)|0.54 (gray)|
>
>    |**SD2.1-Alchemist**|Rel.|Aes.|Comp.|Fidel.|
>    |-|-|-|-|-|
>    |vs SD2.1-original|0.53 (gray)|0.72 (green)|0.85 (green)|0.58 (green)|
>
>    |**SDXL-Alchemist**|Rel.|Aes.|Comp.|Fidel.|
>    |-|-|-|-|-|
>    |vs SDXL-original|0.53 (gray)|0.60 (green)|0.74 (green)|0.57 (green)|
>
>    |**SD3.5M-Alchemist**|Rel.|Aes.|Comp.|Fidel.|
>    |-|-|-|-|-|
>    |vs SD3.5M-original|0.52 (gray)|0.67 (green)|0.74 (green)|0.53 (gray)|
>
>    |**SD3.5L-Alchemist**|Rel.|Aes.|Comp.|Fidel.|
>    |-|-|-|-|-|
>    |vs SD3.5L-original|0.53 (gray)|0.73 (green)|0.80 (green)|0.41 (red)|
>
>    As the results demonstrate, the performance gains from Alchemist fine-tuning are not specific to the PartiPrompts distribution. The Alchemist-tuned models maintain their statistically significant advantage in both Aesthetic Quality and Image Complexity on DrawBench. This provides strong evidence that Alchemist imparts a fundamental improvement to generative quality that generalizes well across different and more challenging prompt scenarios.
>
>    We will add a new subsection to our "Results" section to include this validation.
>
> 4. **On limited model diversity**. We agree with the reviewer that testing across diverse architectures is crucial for validating the "general-purpose" claim. We respectfully argue that although the models tested share the "Stable Diffusion" name, their underlying architectures and training methodologies are sufficiently distinct to provide a strong test of generalizability. Specifically, our selection of five models deliberately spans a wide range of properties:
>
>     - **Architectural Diversity**: The models include both **convolutional U-Net backbones** (SD1.5, SD2.1, SDXL) and a newer **Diffusion Transformer architecture** (SD3.5 models).
>     - **Training Objective Diversity**: The set includes models trained with traditional **diffusion** objectives as well as those trained with **Flow Matching** (SD3.5 models).
>     - **Scale Diversity**: The models vary by an order of magnitude in size, from ~800M parameters (SD1.5) to over 8B parameters (SD3.5 Large).
>     - **Fine-Tuning History Diversity**: The set includes earlier foundation models (presumed to have no SFT) and later models like SD3.5, which, as noted in their paper, already underwent some form of quality fine-tuning.
>
>     The consistent positive impact of Alchemist across this diverse spectrum of architectures, training objectives, scales, and pre-existing optimization levels provides strong evidence for its general-purpose applicability. We will amend the manuscript to more explicitly highlight these dimensions of diversity in our model selection.
>
>     Additionally, recognizing strong community interest in multimodal autoregressive models, we fine-tuned Bagel [2] on the Alchemist dataset. Our side-by-side comparison shows that the Alchemist-tuned Bagel outperforms the original in generating more aesthetic and complex images—though with a slight trade-off in fidelity. We are planning to open-source this checkpoint as well.
>
>     | **Bagel-Alchemist** |Rel.| Aes.| Comp.| Fidel.|
>     |-|-|-|-|-|
>     |vs Bagel-original|0.49 (gray)|0.58 (green)|0.76 (green)|0.42 (red)|
>
> **References**:
>
> [1] Saharia, Chitwan, et al. "Photorealistic text-to-image diffusion models with deep language understanding." Advances in neural information processing systems 35 (2022): 36479-36494.
>
> [2] Deng, Chaorui, et al. "Emerging properties in unified multimodal pretraining." arXiv preprint arXiv:2505.14683 (2025).

---

### Official Review · Reviewer_zbow · 2025-07-02

**Rating:** 5
**Confidence:** 4

**Summary:**

This paper introduces Alchemist, a high-quality SFT dataset for text-to-image models, created by filtering a massive pool of web images using a diffusion model to select high-quality, diverse samples. The authors show that even though the dataset is relatively small—just 3,350 examples—it consistently boosts the aesthetic quality and complexity of images generated by several Stable Diffusion models, without narrowing their diversity. The paper also releases both the dataset and the improved model weights, offering a practical resource for others looking to improve generative model performance with accessible, high-quality data.

**Additional Feedback:**

I really like the topic of this paper. I think it will be more valuable and solid if authors provide more details of the filtering process or open-source the data filtering code.

**Dataset Code Accessibility:**

Yes

**Dataset Code Comments:**

Full dataset and metadata are available.

**Ethical Comments:**

No concern after manual check.

**Ethical Considerations:**

No, there are no or only very minor ethics concerns

**Final Justification:**

The author's rebuttal solves most of my questions. I think it is a good paper exploring high-quality text-to-image dataset construction that should be accepted.

**Limitations Weaknesses:**

Overall, my concerns are relatively minor, but I believe addressing the following points could further strengthen the paper:

- The paper would benefit from providing more detailed information about the data filtering process. Specifically:

    (1) Which watermark detector was used? In my experience, most open-source watermark detectors are quite unreliable, so clarifying this choice is important.

    (2) Why was SIFT-based deduplication chosen over alternatives such as SSCD?

    (3) What is the distribution of the TOPIQ scores in the filtered data, and why was the threshold set at 0.71? From my own experience, most images have TOPIQ scores in the range of 0.4 to 0.8, so a 0.71 threshold seems quite high. How does the chosen threshold affect the diversity of the selected images and does it risk introducing bias?

- Lines 155–159 state that using moderately descriptive prompts for recaptioning leads to better fine-tuning results. Is there any experimental evidence supporting this claim? Because most text-to-image models prefer using longer prompts. Besides, there is a recent paper [1] suggests that longer prompts can reduce text-to-image loss and improve alignment. How do the authors reconcile these findings with their own results?

[1] Qin, Qi, et al. "Lumina-image 2.0: A unified and efficient image generative framework." *arXiv preprint arXiv:2503.21758* (2025).

**Strengths Contributions:**

- I really like the topic of this paper. The community currently lacks systematic explorations into constructing high-quality text-to-image datasets, and this work presents a comprehensive and practical pipeline. Starting from an initial pool of 10B LAION images, the authors progressively filter this to a curated set of 3,350 high-quality samples, which they open-source. This is a significant and meaningful contribution, providing a valuable resource for the community.
- A key strength of the work lies in the novel use of a pre-trained diffusion model as a data quality estimator. As the paper correctly points out, existing image aesthetic and IQA models tend to suffer from substantial bias, often failing to select samples that align well with human. By leveraging internal representations from a pre-trained diffusion model for data filtering, they propose a model-aligned approach without the need for external reward models.
- Models fine-tuned on the Alchemist dataset show clear improvements in both aesthetic quality and image complexity, validating the effectiveness of the proposed dataset and curation method. They also provide insights in terms of SFT, such as conducting ablation studies on dataset size, showing the importance of strict high-quality filtering beyond simply increasing data volume.
- Overall, the paper is well-written, logically organized, and easy to follow.

---

> ### Author Rebuttal · Authors · 2025-07-31
>
> We sincerely thank the reviewer for the feedback and recognition of the importance of the problem under study and experimental contribution. The raised concerns are addressed below:
>
> **Weaknesses**:
>
> 1. **On watermark detector**. We fully agree that the reliability of open-source watermark detectors can be a significant issue. Due to these limitations, we opted to use a proprietary watermark detection model developed internally.
>
>     This model consists of a classification head fine-tuned on a Vision Transformer (ViT) backbone. It was trained on a large-scale, internal dataset containing hundreds of thousands of images, specifically curated to include a highly diverse range of watermarks: semi-transparent logos, text overlays, stock photo watermarks, and tiled patterns at various opacities, scales, and locations.
>
>     To validate its performance, the detector was evaluated on a held-out test set, where it achieves 0.95 accuracy. This high level of accuracy gave us confidence in its ability to effectively filter watermarked content from our initial data pool. We will add a sentence clarifying the nature and performance of this detector in the appendix.
>
> 2. **On SIFT-based deduplication over SSCD**. Initially, SIFT-based deduplication was chosen as a baseline method for its robustness and computational simplicity. While experimenting we prepared and manually examined several SFT datasets (predecessors of Alchemist) for various characteristics, including presence of duplicates. Given that this level of deduplication was sufficient we concluded that the SIFT-based approach provided an excellent trade-off between performance and computational cost for this specific task. We will add a note to the manuscript clarifying this rationale.
>
> 3. **On distribution and threshold of TOPIQ scores**. We confirm the reviewer's observation. The TOPIQ scores of our pre-filtered data form an asymmetric uni-modal distribution peaking around a score of 0.6. Our choice of the stringent 0.71 threshold was therefore a deliberate, data-driven decision, determined empirically by balancing two competing objectives: maximizing technical image quality and preserving broad domain diversity.
>
>     Our ablation studies revealed a clear trade-off. While thresholds higher than 0.71 marginally improved some technical quality metrics, they also introduced significant content bias by disproportionately favoring narrow domains (e.g., architecture and interiors). Conversely, lower thresholds allowed more images with subtle artifacts into the dataset, which we found negatively impacted SFT performance.
>
>     Therefore, the 0.71 threshold represented the optimal sweet spot, effectively removing perceptually flawed images while maintaining the broad thematic coverage necessary for a general-purpose dataset. We will clarify this empirical justification for our threshold selection in the manuscript.
>
> 4. **On using longer prompts**. The relationship between prompt length/style and SFT performance is a critical and nuanced area. To provide direct empirical evidence supporting our claim, we conducted an additional ablation study during the rebuttal period.
>
>     **Experimental Setup**: We created a new version of the Alchemist dataset by re-captioning the 3,350 images using a state-of-the-art long-captioning model, Qwen-VL-Max [1]. As shown in the table below, this resulted in captions that were, on average, significantly longer and more descriptive than our original user-like prompts.
>
>     |                           | Alchemist in-house captioner | Qwen2.5 VL 72B |
>     |---------------------------|-----------------------|----------------|
>     | Average number of symbols | 148                   | 600            |
>     | Average number of words   | 27                    | 100            |
>
>
>     **Fine-Tuning and Evaluation**: We then fine-tuned two representative models, SDXL and SD3.5 Medium, on this new "Alchemist-Long-Caption" dataset and compared their performance against the models fine-tuned on our original Alchemist dataset using the same SbS evaluation protocol and test set.
>
>     **Results**: The results of this comparison are summarized below.
>
>     **Side-by-side comparison**:
>     | Model| Rel. | Aes. | Comp. | Fidel. |
>     |------------------------------|------|------|-------|--------|
>     |  SDXL-Alchemist-QwenVL_caps     |  | | |  |
>     | vs SDXL-original             | 0.55 (green)  | 0.57 (green) | 0.71 (green) | 0.52 (gray)  |
>     | vs SDXL-Alchemist            | 0.52 (gray) | 0.44 (red) | 0.40 (red) | 0.57 (green)  |
>
>     | Model  | Rel. | Aes. | Comp. | Fidel. |
>     |------------------------------|------|------|-------|--------|
>     |  SD3.5M-Alchemist-QwenVL_caps         |  | | |  |
>     | vs SD3.5M-original    | 0.54 (gray)  | 0.61 (green) | 0.63 (green) | 0.50 (gray)  |
>     | vs SD3.5M-Alchemist     | 0.51 (gray) | 0.48 (gray) | 0.44 (red) | 0.46 (gray)  |
>
>     The evaluation clearly demonstrates that while fine-tuning on the long-caption dataset still improves over the baseline, it is less effective at boosting Aesthetic Quality and Image Complexity compared to our original Alchemist dataset with its moderately descriptive, user-like prompts. Interestingly, the longer captions did provide a slight advantage in Fidelity (fewer artifacts), which suggests more descriptive prompts may reduce certain types of errors.
>
>     **Conclusion**: Our findings suggest that there is a trade-off. While longer, more detailed captions may be beneficial for precise alignment and reducing simple defects, moderately descriptive prompts, which more closely mimic typical user interaction, appear to be superior for enhancing the more subjective and crucial qualities of aesthetic appeal. They also enhance visual complexity during SFT. Given that our primary goal with Alchemist was to maximize these perceptual qualities, we selected the user-like prompt style for our final dataset. We will add a summary of this ablation study to the appendix of our revised paper to clarify this important design choice.
>
> **References**:
>
> [1] Bai, Shuai, et al. "Qwen2.5-vl technical report." arXiv preprint arXiv:2502.13923 (2025).

---

> > ### Comment · Reviewer_zbow · 2025-08-01
> >
> > thanks for the rebuttal, I have raised my score.

---

### Comment · Area_Chair_dFAp · 2025-08-06
**Please engage in the reviewer-author discussion.**

Dear Reviewers,

Thank you for providing the initial reviews. Please respond to the authors' rebuttal and engage in the reviewer-author discussion if you haven't done so.

Best,

AC

---

### Note · Authors · 2025-08-12

We sincerely thank the reviewers for their thorough and constructive feedback. Their insightful questions have enabled us to significantly strengthen the clarity, rigor, and overall contribution of our work. In response, we have made substantial revisions, primarily centered on three key areas.

First, we have provided **detailed clarifications on our data curation methodology**. This includes justifying our choice of a high-performance proprietary watermark detector over unreliable open-source alternatives, our use of a SIFT-based deduplication method (validated empirically for its efficiency), and the data-driven process for selecting the 0.71 TOPIQ threshold to optimally balance quality and diversity. We have also expanded on our prompt and calibration set design, detailing the steps taken to mitigate potential stylistic biases in our novel diffusion-based estimator.

Second, in direct response to the reviews, we **conducted several new experiments to strengthen our claims of generalizability and robustness**. We now include an evaluation on the DrawBench benchmark, confirming that Alchemist's benefits extend to different prompt distributions. We have added a quantitative analysis of generative diversity and new ablation studies on both caption style (validating our choice of moderately descriptive prompts over longer ones) and smaller dataset sizes (confirming that 3,350 samples is a robust sweet spot).

Finally, we have **enhanced the transparency and clarity of our evaluation protocol**. We have clarified that our human evaluation involved a large pool of over 1,000 expert annotators (with three assigned per-task) to ensure robustness. We have also performed a new VLM-as-a-judge evaluation with GPT-4o, supporting our human findings. To further aid reproducibility, we have committed to releasing our 1k-image calibration set, pending a final ethical review.

We are confident that these revisions, particularly the new experimental evidence, have substantially improved the manuscript and more rigorously validated the contributions of Alchemist. We thank the reviewers again for their valuable guidance.

---

### Decision · Program_Chairs · 2025-09-18

**Decision:**

Accept (poster)

**Comment:**

This paper proposes novel dataset curation methodology that leverages pre-trained diffusion models as quality estimators, enabling the creation of Alchemist, a compact yet effective supervised fine-tuning dataset for text-to-image models. The multi-stage filtering pipeline, particularly the diffusion-based estimator, provides a model-aligned and bias-resistant approach to selecting high-quality samples. Experiments, including both automated metrics and human evaluations, demonstrate clear improvements in generative quality, aesthetics, and complexity across multiple public models, confirming the dataset’s effectiveness. The paper has good clarity, transparency, reproducibility, and open-sourcing of data and models, making it a valuable resource for the community. Overall, this is a well-executed and timely contribution that addresses a critical gap in building general-purpose SFT datasets for T2I models.